# How Flexible Is the Concept of Local Thermodynamic Equilibrium?

**DOI:** 10.3390/e25010145

**Published:** 2023-01-10

**Authors:** Vijay M. Tangde, Anil A. Bhalekar

**Affiliations:** 1Department of Chemistry, Rashtrasant Tukadoji Maharaj Nagpur University, Nagpur 440 033, India; 2106, Himalaya Prestige, South Ambazari Marg, SBI Colony, Gopal Nagar, Nagpur 440 022, India

**Keywords:** local thermodynamic equilibrium, classical irreversible thermodynamics, thermodynamic state functions in nonequilibrium, Gibbs relations

## Abstract

It has been demonstrated by using generalized phenomenological irreversible thermodynamic theory (GPITT) that by replacing the conventional composition variables $\left\{x_{k}\right\}$ by the quantum level composition variables $\left\{{\tilde{x}}_{k,\;j}\right\}$ corresponding to the nonequilibrium population of the quantum states, the resultant description remains well within the local thermodynamic equilibrium (LTE) domain. The next attempt is to replace the quantum level composition variables by their respective macroscopic manifestations as variables. For example, these manifestations are, say, the observance of fluorescence and phosphorescence, existence of physical fluxes, and ability to register various spectra (microwave, IR, UV-VIS, ESR, NMR, etc.). This exercise results in a framework that resembles with the thermodynamics with internal variables (TIV), which too is obtained as a framework within the LTE domain. This TIV-type framework is easily transformed to an extended irreversible thermodynamics (EIT) type framework, which uses physical fluxes as additional variables. The GPITT in EIT version is also obtained well within the LTE domain. Thus, GPITT becomes a complete version of classical irreversible thermodynamics (CIT). It is demonstrated that LTE is much more flexible than what CIT impresses upon. This conclusion is based on the realization that the spatial uniformity for each tiny pocket (cell) of a spatially non-uniform system remains intact while developing GPITT and obviously in its other versions.

## 1. Introduction

In this paper, for the sake of simplicity, we used a non-uniform single phase multi-component system with chemical reactions at non-vanishing rates. This is because our main aim, as the title of the paper indicates, is to illustrate the contents of the local thermodynamic equilibrium (LTE) (see, for example, [1,2]), which is one of the basic ingredients of the classical irreversible thermodynamics (CIT) [1,2,3,4,5,6,7]. A pictorial representation of LTE is depicted in Figure 1.

That is, a spatially non-uniform system is conceptually divided in each spatially uniform tiny volume element, say, of mass δm, and the seat of gradients and physical fluxes lie across the boundaries of each one of them. Next, it is assumed that the local time rate version of the Gibbs relation of equilibrium thermodynamics is the complete operating description of LTE that reads as
(1)$$\begin{array}{c} \begin{array}{c} T\frac{ds}{dt}=\frac{du}{dt}+p\frac{dv}{dt}-\sum_{k}\mu_{k}\frac{dx_{k}}{dt} \end{array} \end{array}$$
where *s* and *u*, respectively, are the per unit mass entropy and internal energy, *v* is the specific volume, $\left\{x_{k}\right\}$ are the mass fractions of the components within the system, $\mu_{k}$ is the local chemical potential per unit mass of the component *k*, *T* is the local temperature, *p* is the local pressure, and *t* is time. As Equation (Equation 1) is the per unit mass version of the Gibbs relation derived in equilibrium thermodynamics based on the first, second and zeroth law of thermodynamics, it is assumed that all the functions in it have the same thermodynamic status as that in equilibrium thermodynamics.

Thus, in place of Equation (Equation 1), an equivalent prescription of LTE reads as
(2)$$\begin{array}{c} \begin{array}{c} s\left(r,t\right)=s\left(u\left(r,t\right),v\left(r,t\right),\{x_{k}\left(r,t\right)\}\right) \end{array} \end{array}$$
where r is the positional coordinate.

Thereafter, the following entropy balance equation, the Clausius–Duhem inequality, follows: (3)$$\begin{array}{c} \begin{array}{c} \rho \frac{ds}{dt}+\nabla \cdot J_{s}=\sigma_{s}>0 \end{array} \end{array}$$
where ρ is the mass density, and the expressions of entropy flux density $J_{s}$ and the entropy source strength $\sigma_{s}$ read as follows: (4)$$\begin{array}{c} \begin{array}{c} J_{s}=\frac{q-\sum_{k}\mu_{k}J_{k}}{T} \end{array} \end{array}$$
and
(5)$$\begin{array}{c} \begin{array}{c} \sigma_{s}=q\cdot \nabla \left(\frac{1}{T}\right)+\frac{1}{T}\Pi :\nabla u-\frac{1}{T}\sum_{k}J_{k}\cdot \left(T\nabla \left(\frac{\mu_{k}}{T}\right)-F_{k}\right)+\sum_{\alpha }\frac{A_{\alpha }}{T}\frac{d\xi_{\alpha }}{dt}>0 \end{array} \end{array}$$
where q is the so-called heat flux density, Π is the dissipative stress tensor, u is the barycentric velocity vector, $F_{k}$ are the conservative body forces, $J_{k}$ are the respective diffusion flux densities, $A_{\alpha }$ is the chemical affinity of the α-th chemical reaction defined below in Equation (Equation 8), and $\xi_{\alpha }$ is the extent of the advancement of the α-th chemical reaction. In arriving at the above expressions, the following internal energy balance equation,
(6)$$\begin{array}{c} \begin{array}{c} \rho \frac{du}{dt}=-\nabla \cdot q-p\rho \frac{dv}{dt}+\Pi :\nabla u+\sum_{k}J_{k}\cdot F_{k}, \end{array} \end{array}$$
and the mass balance equations,
(7)$$\begin{array}{c} \begin{array}{c} \;\rho \frac{dx_{k}}{dt}=-\nabla \cdot J_{k}+\sum_{\alpha }\nu_{k}^{\alpha }M_{k}\frac{d\xi_{\alpha }}{dt} \end{array} \end{array}$$
were used.

The chemical affinity has the following standard definition: (8)$$\begin{array}{c} \begin{array}{c} A_{\alpha }=-\sum_{k}\nu_{k}^{\alpha }M_{k}\mu_{k}. \end{array} \end{array}$$

In Equation (Equation 8), $\nu_{k}^{\alpha }$ is the stoichiometric coefficient of the component *k* in α-th chemical reaction and, by convention, is taken as a positive number for the products and negative number for the reactants, and $M_{k}$ is the molar mass of the component *k*.

The great achievement of CIT lies in the fact that innumerable experimentally established phenomena right from the 17th century for the first time obtained the thermodynamic base. In this, Onsager relations [10,11] played a major role. Even the nonlinear flux-force relationships obtained the thermodynamic base in the domain of CIT (see for example [12]). Recall also that the various softwares developed in the field of computational thermodynamics (CT) are based on the LTE assumption. In doing so the general thermodynamic work of Hillert has been used as a basic building block which too has the LTE base (refer, for example, to [13,14,15,16,17,18,19,20,21,22]).

The present discussion would illustrate that the spatial uniformity of the conceived tiny volume elements is the primary aspect of LTE, which is vindicated by the tremendous success of CIT.

The belief that there are several irreversible processes those cannot be accommodated in the domain of LTE led workers to develop new thermodynamic frameworks. These proposals were lucidly summarized in [23]. The notable ones amongst them, from the point of view of the present discussion, are extended irreversible thermodynamics (EIT) (see for example [24,25,26,27,28,29]), thermodynamics with internal variables (TIV) (see for example [30,31,32,33,34,35,36,37,38,39]), Keizer’s version of nonequilibrium thermodynamics which is based on the fluctuations about nonequilibrium stationary states with incorporation of imbalances between the respective rates of forward and reverse elementary process [40,41,42,43,44,45,46,47,48,49] and the thermo-kinetics of Oláh et al. [50], which assumes that the operating thermodynamic forces and their fluxes are the result of an imbalance between their respective forward and reverse components. In these frameworks, except the last one, additional thermodynamic variables are introduced (in Keizer’s version, there also appear reservoir intensities as additional variables) and obviously, it is considered that it amounts to go beyond the LTE domain. Hence, one is led to the concept of nonequilibrium entropy (say η or Σ), nonequilibrium temperature θ or $\hat{T}$ and nonequilibrium pressure π or $\hat{p}$ as physically different quantities from the corresponding local equilibrium ones: *s*, *T* and *p*. Recently, a novel definition of nonequilibrium temperature has been proposed based on the Gouy–Stodola and Carnot theorems [51]. In CIT, the thermodynamic intensities *T* and *p* are coincided with the experimentally measured ones, as is the case in equilibrium thermodynamics.

In the present discussion, our aim is to dwell over the prevailing belief that the incorporation of additional variables to the functional dependence shown in Equation (Equation 2) amounts to the breakdown of LTE. For this purpose, we considered the relevant basic features of the generalized phenomenological irreversible thermodynamic theory (GPITT) [52,53,54]. It illustrates that the breakdown of LTE does not follow, even if one uses the physical fluxes and/or the hidden internal variables as additional thermodynamic variables, provided their physical existence is traced out in corresponding elementary processes. In other words, we arrived at a version of GPITT that resembles the framework known as TIV [30,31,32,33,34,35,36,37,38,39], which is believed to be a description beyond the LTE domain. However, the GPITT version that resembles TIV is obtained well within the LTE domain. We further show that the GPITT version that resembles EIT is a special case of the GPITT version of TIV and hence it too belongs to the LTE domain. We also discussed how the thermodynamics description using the GPITT framework works using various constitutive equations, particularly of the physical fluxes coupled with the nonequilibrium population of translational quantum states. We identified two different sets of thermodynamic variables: one belongs to the fast domain of time (the additional thermodynamic variables) and the other set of variables are those that appear in the conventional thermodynamics and belong to the slow domain of time.

## 2. Basic Approach Leading to the Generalized Phenomenological Irreversible Thermodynamic Theory

Herein we summarize the basics of the approach adopted while developing the GPITT [52,53,54]. It is based on the fact that the existence of physical fluxes traces its origin in the nonequilibrium population of translational quantum states. In the earlier attempt [52,53,54], we argued that for an ideal gas, if the molecular distribution of peculiar or chaotic velocity corresponds to the Maxwellian (that is equilibrium population of translational quantum states), we have q=0 and Π=0. If the said distribution function is non-Maxwellian (that is, we have a nonequilibrium population of translational quantum states), we have q≠0 and Π≠0. This fact directed us to replace the standard composition variables $\left\{x_{k}\right\}$ appearing in Equation (Equation 1) by $\left\{{\tilde{x}}_{k,\;j}\right\}$, the set of mass fraction of the components identified by the variable subscript *k* in the quantum state *j* in the case of the nonequilibrium population of quantum states, that leads to the current version of GPITT. Therefore, the appropriate form of the Gibbs relation instead of that in Equation (Equation 1) reads as
(9)$$\begin{array}{c} \begin{array}{c} T\frac{ds}{dt}=\frac{du}{dt}+p\frac{dv}{dt}-\sum_{k,\;j}{\tilde{\mu }}_{k,\;j}\;\frac{d\;{\tilde{x}}_{k,\;j}}{dt} \end{array} \end{array}$$The need to distinguish between $\left\{x_{k,\;j}\right\}$ and $\left\{{\tilde{x}}_{k,\;j}\right\}$ lies in the fact that the equilibrium population of quantum states is represented by $\left\{x_{k,\;j}\right\}$, which numerically is not identically same with that in the nonequilibrium. That is, there, we have $x_{k,\;j}\neq {\tilde{x}}_{k,\;j}$, though we do have
(10)$$\begin{array}{c} \begin{array}{c} \sum_{j}x_{k,\;j}=x_{k}=\sum_{j}{\tilde{x}}_{k,\;j} \end{array} \end{array}$$
where the subscript *j* represents the *j*-th quantum state. Recall that, in the kinetic theory of non-uniform gases, the velocity distribution function *f* is computed by an expression $f=f^{Maxwell}\left(1+\Phi \right)$, where Φ is the nonequilibrium contribution, which in the case of spatial uniformity vanishes [55]. The distribution function determines the number density of molecules with a particular velocity (chaotic) that leads us to use in its place the corresponding mass fractions. There are certain studies in which the nonequilibrium distribution of molecular velocity corresponding was computed and compared with the corresponding Maxwellian or Guassian distribution. This distinction of the distribution functions is depicted in Figure 2 and Figure 3. In Figure 2, the velocity distribution is depicted for the Maxwell (corresponds to spatial uniformity), and that on using the Grad 13 moment and the direct simulation Monte Carlo (DSMC) methods for the nonequilibrium distributions are computed and depicted for the y component of the velocities. The three curves of Figure 2 clearly show the said difference. In another approach (see Figure 3), presented in [56], are the plots of redistribution R defined as R = $P_{S}-P_{v}$, where $P_{S}$ is the true distribution and $P_{v}$ is the Gaussian. Notice the large difference between $P_{v}$ and $P_{S}$ when depicted as distribution R, whereas the said distinction is very weakly depicted on plotting vs. distribution in two cases.

In view of the above fact, the mass balance law is amended to the following (the Boltzmann integro-differential equation [55,57] also describes the same conservation law, but in terms of distribution function; however, the details of it would be described separately):
(11)$$\begin{array}{c} \begin{array}{c} \rho \;\frac{d\;{\tilde{x}}_{k,\;j}}{dt}\;=\;-\;\nabla \cdot {\tilde{J}}_{k,\;j}\;+\;\nu_{k,\;j}\;M_{k}\;\frac{d\omega }{dt}\;+\;\sum_{\alpha }\nu_{k}^{\alpha }M_{k}{\tilde{\gamma }}_{{}_{k,\;j}}\;\frac{d\xi_{\alpha }}{dt}, \end{array} \end{array}$$
where ρ’s are the respective mass densities, ${\tilde{x}}_{{}_{k,\;j}}={\tilde{\rho }}_{{}_{k,\;j}}/\rho$ is the nonequilibrium mass fraction in the quantum state *j* of the component *k*, ${\tilde{\gamma }}_{{}_{k,\;j}}={\tilde{\rho }}_{{}_{k,\;j}}/\rho_{k}$, the component-wise nonequilibrium mass fraction in the quantum state *j*, $\nu_{k,\;j}$ is the stoichiometric coefficient of the component *k* in the collisional mechanism of the population equilibration process for the *j*-th quantum state, and ω is the extent of advancement of population equilibration in internal molecular quantum states (that is, we identified an additional irreversible process of scalar nature, which is the population equilibration of quantum states by molecular collisions). If it is considered that there we have the nonequilibrium population of only translation, rotation, vibration and electronic quantum states, then the energy, ε, of the *j*-th quantum state is given by
(12)$$\begin{array}{c} \begin{array}{c} \varepsilon_{j}=\varepsilon_{n_{j}}+\varepsilon_{J_{j}}+\varepsilon_{v_{j}}+\sum_{\mathrm{all}\;\mathrm{electronic}\;\mathrm{states}}\varepsilon_{elec} \end{array} \end{array}$$
where the subscripts $n_{j}$, $v_{j}$ and $J_{j}$ are the translational, vibrational and rotational quantum numbers of the overall quantum state *j*.

**Figure 2 entropy-25-00145-f002:**
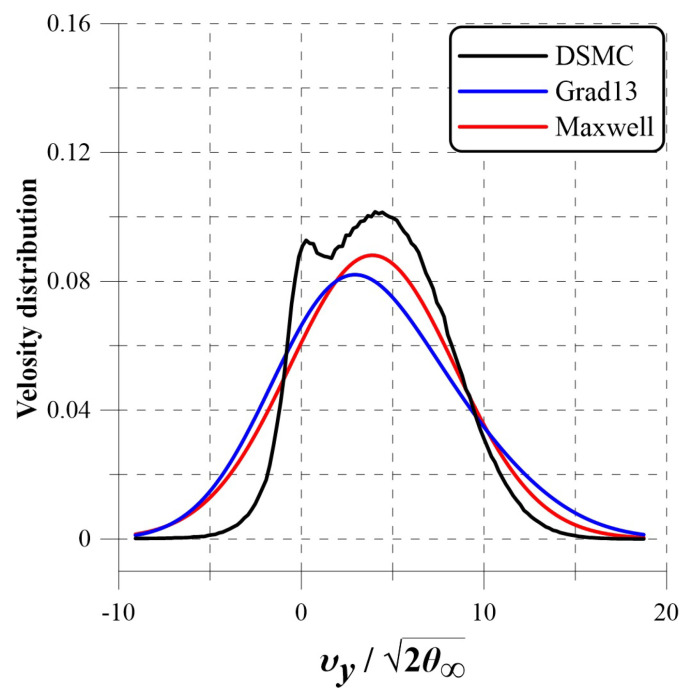
The comparison of Maxwell, Grad and direct simulation Monte Carlo (DSMC) computations distribution function of z component of velocity [58]. The last two are the corresponding nonequilibrium velocity distribution functions. In this figure, θ is the temperature in energy units, and the peculiar velocity is normalized by the average free-stream molecular velocity is $\sqrt{2\theta_{\infty }}$.

**Figure 3 entropy-25-00145-f003:**
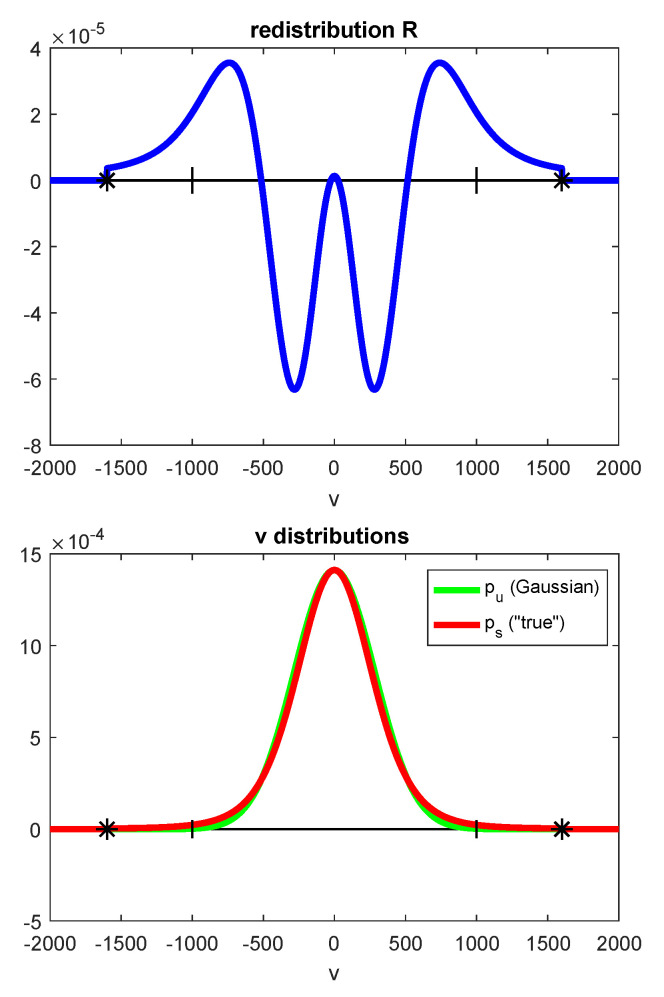
The schematic representation of the comparison of the proposed Gaussian distribution of velocities and the corresponding actual velocity distribution in the bottom plot where the deviation is extremely small. The same is depicted using the redistribution function R defined as R = $P_{S}-P_{v}$, where $P_{S}$ is the true distribution, and $P_{v}$ is the Gaussian. The plots were taken from [56] with the permission of the corresponding author.

Thus, notice that the second term on the r.h.s. of Equation (Equation 11) corresponds to an irreversible process so far not identified but that exists whenever quantum states have a nonequilibrium population. However, this mechanism is eliminated from our consideration on summing the expression of Equation (Equation 11) over all quantum states, because by definition, we have,
(13)$$\begin{array}{c} \begin{array}{c} \sum_{j}\nu_{k,\;j}=0 \end{array} \end{array}$$This is so because when a quantum state has a smaller population than when in equilibrium, then by collisions, its population increases, and hence, correspondingly, $\nu_{k,\;j}$ have +1 value and, for those quantum states, have more population than the corresponding equilibrium one, for which we have $\nu_{k,\;j}=-1$. In each collision binary or even higher ones, the number of $\nu_{k,\;j}=+1$ is always equal to the number with $\nu_{k,\;j}=-1$. In this way, summing the expression of Equation (Equation 11) over all quantum states reduces to the standard mass balance expression of Equation (Equation 7). This illustrates the need to use Equation (Equation 11) instead of Equation (Equation 7) if one wishes to retain in our consideration this internal mechanism of irreversibility.

In thermodynamic language, when the populations of quantum states correspond to the nonequilibrium, we have
(14)$$\begin{array}{c} \begin{array}{c} {\tilde{\mu }}_{k,\;j}\neq {\tilde{\mu }}_{k,\;j^{\prime }}\neq {\tilde{\mu }}_{k,\;j^{\prime \prime }}\neq \cdots \cdots \cdots \neq {\tilde{\mu }}_{k} \end{array} \end{array}$$
and corresponding to the equilibrium population, we have
(15)$$\begin{array}{c} \begin{array}{c} {\tilde{\mu }}_{k,\;j}=\mu_{k,\;j},\;\mu_{k,\;j}=\mu_{k,\;j^{\prime }}=\mu_{k,\;j^{\prime \prime }}=\cdot \cdot \cdot \cdot =\mu_{k} \end{array} \end{array}$$In this way, the Gibbs relation of Equation (Equation 9) is reduced to that of Equation (Equation 1) because ${\tilde{x}}_{{}_{k,\;j}}⟶x_{k,\;j}$ under the condition of Equation (Equation 15). However, for example, no physical fluxes are allowed to exist. Then, one is not permitted to combine Equation (Equation 1) with Equations (Equation 6) and (Equation 7). In the case of the nonequilibrium population of quantum states, ${\tilde{\mu }}_{k}$ is computed as
(16)$$\begin{array}{c} \begin{array}{c} {\tilde{\mu }}_{k}=\sum_{j}{\tilde{\gamma }}_{{}_{k,\;j}}\;{\tilde{\mu }}_{k,\;j} \end{array} \end{array}$$Thus, the use of ${\tilde{\mu }}_{k}$, that is, $\;\tilde{}\;$ over the symbol of the chemical potential reminds us that it corresponds to when we have the nonequilibrium population of quantum states.

Hence, it is clear that the replacement of $\sum_{k}\mu_{k}\frac{dx_{k}}{dt}$ appearing in Equation (Equation 1) by the term $\sum_{k,\;j}{\tilde{\mu }}_{k,\;j}\frac{d\;{\tilde{x}}_{k,\;j}}{dt}$ does not imply the breakdown of LTE. Recall that it is just the case of choosing a different set of composition variables already existing in the system. It is just a case of accounting for the existing source of irreversibility, which in CIT, remains unaccounted for. This gets best reflected in the expression of the entropy source strength of GPITT obtained by using Equations (Equation 6) and (Equation 11), that reads as
(17)$$\begin{array}{c} \begin{array}{c} \begin{array}{c} \sigma_{s}=q\cdot \nabla \left(\frac{1}{T}\right)+\frac{1}{T}\Pi :\nabla u-\frac{1}{T}\sum_{k}J_{k}\cdot \left(T\nabla \left(\frac{{\tilde{\mu }}_{k}}{T}\right)-F_{k}\right)\\ +\sum_{\alpha }\frac{{\tilde{A}}_{\alpha }}{T}\frac{d\xi_{\alpha }}{dt}+\frac{1}{T}\sum_{k,\;j}\;{\tilde{\mu }}_{k,\;j}\;J_{k}\cdot \nabla \left({\tilde{\gamma }}_{{}_{k,\;j}}\right)+\frac{\tilde{B}}{T}\frac{d\omega }{dt}>0 \end{array} \end{array} \end{array}$$Notice that each term of Equation (Equation 17) is a product of flux and its driving force. In Figure 4, for the sake of illustration, we have depicted in one dimension of the processes taking place within and across a tiny volume element of a non-uniform fluid and variation of intensive properties on account of existing gradients. $J_{i,x}$ is the flux density of the *i*-th process in the *x*-direction and $X_{i}$ is the *i*-th intensive property. Moreover, there are two last terms on the r.h.s. of Equation (Equation 17) but do not appear in Equation (Equation 5) the latter equation in CIT is understood to be valid for a system with existence of physical fluxes. These three terms of Equation (Equation 17) along with the use of ${\tilde{\mu }}_{k}$ and ${\tilde{A}}_{\alpha }$ take care of the irreversibility associated with the nonequilibrium population of the molecular quantum states. Where the expression of $\tilde{B}$, the *internal population equilibration affinity* through the collisional mechanism reads as
(18)$$\begin{array}{c} \begin{array}{c} \tilde{B}\;=\;-\;\sum_{{}_{k,\;j}}\;{\tilde{\mu }}_{k,\;j}\;\nu_{k,\;j}M_{k} \end{array} \end{array}$$
and that of the chemical affinity of the α-th chemical reaction is defined as
(19)$$\begin{array}{c} \begin{array}{c} {\tilde{A}}_{\alpha }\;=\;-\sum_{k,\;j}\;\nu_{k}^{\alpha }M_{k}{\tilde{\gamma }}_{k,\;j}\;{\tilde{\mu }}_{k,\;j}=-\sum_{k}\;\nu_{k}^{\alpha }M_{k}{\tilde{\mu }}_{k}, \end{array} \end{array}$$It amounts to amending the functional dependence of Equation (Equation 2). The corrected one reads as
(20)$$\begin{array}{c} \begin{array}{c} s\left(r,t\right)=s\left(u\left(r,t\right),v\left(r,t\right),\{{\tilde{x}}_{k,\;j}\left(r,t\right)\}\right) \end{array} \end{array}$$The corresponding equations of state read as
(21)$$\begin{array}{c} \begin{array}{c} {\left(\frac{\partial s}{\partial u}\right)}_{v,\;\tilde{x}}=\frac{1}{T}\;,\;{\left(\frac{\partial s}{\partial v}\right)}_{u,\;\tilde{x}}=\frac{p}{T}\;,\;{\left(\frac{\partial s}{\partial {\tilde{x}}_{k,\;j}}\right)}_{v,\;{\tilde{x}}^{\prime }}=-\frac{{\tilde{\mu }}_{k,\;j}}{T} \end{array} \end{array}$$
where the subscript $\tilde{x}$ denotes all composition variables kept constant, and the subscript ${\tilde{x}}^{\prime }$ denotes, except ${\tilde{x}}_{k,\;j}$, all the composition variables kept constant.

Since the transformation of the functional dependence of Equation (Equation 2) to that given in Equation (Equation 20) is the case of replacing the macroscopic level of composition variables by those that do exist but are expressed at the quantum level, there is no reason to expect the breakdown of LTE. It once again demonstrates that all the intensities appearing in GPITT belong to the LTE domain.

In this way, the per unit mass Gibbs function G has the following expression:(22)$$\begin{array}{c} \begin{array}{c} G=\sum_{k,\;j}\;{\tilde{x}}_{k,\;j}\;{\tilde{\mu }}_{k,\;j} \end{array} \end{array}$$
correspondingly, the functional dependence of the Gibbs function in GPITT reads as
(23)$$\begin{array}{c} \begin{array}{c} G=G\left(T,\;p,\;\{{\tilde{x}}_{k,\;j}\}\right) \end{array} \end{array}$$

The above discussion illustrates that the conventional treatment of CIT is ignorant of the existing sources of irreversibility due to the nonequilibrium population of quantum states because it directly uses the mass balance expression of Equation (Equation 7), which, as it is, is not capable of accounting for the irreversible processes associated with the nonequilibrium population of quantum states.

It is interesting to see that the expressions of Equation (Equation 22) are transformed as follows: (24)$$\begin{array}{c} \begin{array}{c} G=\sum_{k,\;j}\;{\tilde{x}}_{k,\;j}\;{\tilde{\mu }}_{k,\;j}=\sum_{k,\;j}\;x_{k}\times {\tilde{\gamma }}_{{}_{k,\;j}}\;{\tilde{\mu }}_{k,\;j}=\sum_{k}\;x_{k}\;{\tilde{\mu }}_{k} \end{array} \end{array}$$
where we adopted the definition $\sum_{j}{\tilde{\gamma }}_{{}_{k,\;j}}{\tilde{\mu }}_{k,\;j}={\tilde{\mu }}_{k}$ and used the basic identities ${\tilde{x}}_{k,\;j}={\tilde{\gamma }}_{{}_{k,\;j}}x_{k}$ and ${\tilde{\gamma }}_{{}_{k,\;j}}={\tilde{\rho }}_{{}_{k,\;j}}/\rho_{k}$. Then, by substitution of the last summational term on the r.h.s. of Equation (Equation 24) for G in the standard expression of the Gibbs function $Ts=u+pv-G=u+pv-\sum_{k}\;x_{k}\;{\tilde{\mu }}_{k}$, the resulting Gibbs relation reads as: (25)$$\begin{array}{c} \begin{array}{c} T\frac{ds}{dt}=\frac{du}{dt}+p\frac{dv}{dt}-\sum_{k}{\tilde{\mu }}_{k}\frac{dx_{k}}{dt} \end{array} \end{array}$$
and the corresponding expressions of $\sigma_{s}$ and $J_{s}$ read as
(26)$$\begin{array}{c} \begin{array}{c} \sigma_{s}=q\cdot \nabla \left(\frac{1}{T}\right)+\frac{1}{T}\Pi :\nabla u-\frac{1}{T}\sum_{k}J_{k}\cdot \left(T\nabla \left(\frac{{\tilde{\mu }}_{k}}{T}\right)-F_{k}\right)+\sum_{\alpha }\frac{{\tilde{A}}_{\alpha }}{T}\frac{d\xi_{\alpha }}{dt}\geq 0 \end{array} \end{array}$$
and $J_{s}$ is the entropy flux density given by
(27)$$\begin{array}{c} \begin{array}{c} J_{s}=\frac{q-\sum_{k}{\tilde{\mu }}_{k}J_{k}}{T} \end{array} \end{array}$$

The Gibbs–Duhem equation that accompanies the Gibbs relation of Equation (Equation 25) reads as
(28)$$\begin{array}{c} \begin{array}{c} s\frac{dT}{dt}-v\frac{dp}{dt}+\sum_{k}x_{k}\frac{d\;{\tilde{\mu }}_{k}}{dt}=0 \end{array} \end{array}$$

Notice that we do not have ${\tilde{\mu }}_{k}=\mu_{k}$ because of the the non-equivalence $\sum_{k,\;j}{\tilde{\mu }}_{k,\;j}{\tilde{\gamma }}_{{}_{k,\;j}}\neq \sum_{k,\;j}\mu_{k,\;j}\gamma_{{}_{k,\;j}}$. That is, the use of ${\tilde{\mu }}_{k,\;j}$ and ${\tilde{\gamma }}_{{}_{k,\;j}}$ implies the existence of the nonequilibrium population of quantum states, and hence the existence of gradients and corresponding fluxes are implied. In contrast, the use of $\mu_{k,\;j}$ and $\gamma_{{}_{k,\;j}}$ is valid when the fluxes and gradients do not exist. Additionally, notice the difference in the two expressions of entropy source strength of Equations (Equation 5) and (Equation 26). In the latter, there appear the terms ${\tilde{\mu }}_{k}$ and ${\tilde{A}}_{\alpha }$ that confirm the existence of fluxes and gradients. However, on comparing Equations (Equation 17) and (Equation 26), we find that for the last two terms on the r.h.s., the former expressions are missing in the latter one. It then gives an impression that the corresponding mechanisms of irreversibility are unaccounted for by the latter expression. Hence, it seems that we have landed in the incomplete description of LTE. However, in Section 3.2.2, we will see that this expression is accompanied by the operation of the functional dependencies $q=q(u,\;v,\;\left\{x_{k}\right\})$, $\Pi =\Pi (u,\;v,\;\left\{x_{k}\right\})$ and for all $J_{k}=J_{k}\left(u,\;v,\;\left\{x_{l}\right\}\right)$; this is because the description is relegated to the *slow domain of time* that we have identified therein.

This further illustrates that the conventional method of CIT is ignorant of the existing sources of irreversibility due to the nonequilibrium population of quantum states; hence, therein one uses the mass balance expression of Equation (Equation 7), which is not capable of accounting for the irreversible processes associated with the nonequilibrium population of quantum states.

Although the above described structure of GPITT takes care of all the possible sources of irreversibility, it also uses the parameters defined at the quantum level. In the next Section 3, we describe our investigations that results in a GPITT description in terms of all macroscopic level of variables. The GPITT framework that results resembles that of EIT. The above discussion answers indirectly in negation of our query regarding the breaking down of LTE. This is indirect because the replacement of the conventional variables, ($\left\{x_{k}\right\}$), by the variables ($\left\{{\tilde{x}}_{k,\;j}\right\}$) defined at the quantum level, does amount to a tremendous increase in the number of variables. It demonstrates that such a large increase in the number of variables compared to those provided by the equilibrium thermodynamics does not lead to the breakdown of LTE. However, it is indeed a case of replacement of the conventional set of composition variables by another set of composition variables defined at the quantum level, but nature wise there is no change. Therefore, it would be better if we are able to breakdown these latter variables into a set of corresponding macroscopic ones. Fortunately, it is possible to do so. This task we have undertaken and is the subject matter of the next Section 3.

## 3. Transforming GPITT to All Macroscopic Variables

In order to remove the above difficulty, we recall that the global level manifestation of the nonequilibrium population of translational quantum states is the existence of physical fluxes. Moreover, the ability to record microwave, IR, UV-VIS, NMR and ESR spectra is the global level manifestation of the nonequilibrium population of corresponding quantum states and the observation of fluorescence and phosphorescence is the global level manifestation of the nonequilibrium population of singlet and triplet excited states. The illustration of how fluorescence and phosphorescence originate at the global level is schematically depicted by the Jablonski diagram of Figure 5.

It is profitable to first recall some relevant aspects of De Donderian chemical thermodynamics [59]. For a spatially uniform closed system undergoing a single chemical reaction at a non-vanishing rate, the De Donderian equation reads as
(29)$$\begin{array}{c} \begin{array}{c} T\frac{dS}{dt}=\frac{dU}{dt}+p\frac{dV}{dt}+A\frac{d\xi }{dt} \end{array} \end{array}$$

The symbols in Equation (Equation 29) have standard meanings. Thus, we see that ξ is an internal variable because it does not help in quantifying the exchange of energy/entropy. Next, it is easy to transform the Gibbs relation of Equation (Equation 25) to,
(30)$$\begin{array}{c} \begin{array}{c} T\frac{ds}{dt}=\frac{du}{dt}+p\frac{dv}{dt}-\sum_{k}{\tilde{\mu }}_{k}\frac{d_{e}x_{k}}{dt}+\tilde{A}\;\frac{d\xi }{dt} \end{array} \end{array}$$

Notice that in arriving at Equation (Equation 30), the following splitting is affected: (31)$$\begin{array}{c} \begin{array}{c} -\sum_{k}{\tilde{\mu }}_{k}\frac{dx_{k}}{dt}=-\sum_{k}{\tilde{\mu }}_{k}\frac{d_{e}x_{k}}{dt}+\tilde{A}\;\frac{d\xi }{dt} \end{array} \end{array}$$
where the first term on its r.h.s. is the exchange term, whereas the second one describes the internal process. As stated above, ξ is the internal variable whose physical meaning is well understood. On the same lines, we wish to split $-\sum_{k,\;j}{\tilde{\mu }}_{k,\;j}\frac{d{\tilde{x}}_{k,\;j}}{dt}$ into the number of terms describing each existing internal process. This can be achieved in two ways described in the following Section 3.1 and Section 3.2.

Herein, it is interesting to recall that in the introduction of the textbook entitled Chemical Thermodynamics by Prigogine and Defay [59], it is asserted that “A Thermodynamics of chemical reactions must necessarily be a thermodynamics of irreversible phenomena”. This is why the De Donderian Equation (Equation 30) is in the time rate form.

### 3.1. Gibbs Relation of GPITT with Internal Variables as Fast Variables

Recall that, in photochemical kinetics $S_{0}$, $S_{1}$, $S_{2}$, … and $T_{0}$ (an exceptional example is that of Oxygen molecule whose triplate state is the ground state), $T_{1}$, $T_{2}$, …… are treated as separate chemical species [60,61,62]. Hence, the processes shown in Figure 6 that is $S_{1}\to S_{0}$, $S_{2}\to S_{1}$, $S_{1}\to T_{1}$, $T_{1}\to S_{0}$. Other processes originating in the nonequilibrium population of quantum states are the conversion between nuclear spin states (such and *o*- and *p*- hydrogen molecules), relaxation back between nuclear (NMR) and electron (ESR) spin states, rotation–vibration relaxations (microwave and IR spectroscopy), the time variation of physical fluxes, etc., all of them contributing to the time variation of ${\tilde{x}}_{k,\;j}$. Thus, they are identified as internal processes similar to the rate of chemical reactions. Moreover, the chemical reactions of electronic excited states also fall within the domain of the *fast domain of time* (see, for example, Figure 7).

Herein, the Gibbs relation of Equation (Equation 9) on the lines of Equation (Equation 30) are expressed as,
(32)$$\begin{array}{c} \begin{array}{c} T\frac{ds}{dt}=\frac{du}{dt}+p\frac{dv}{dt}-\sum_{k}{\tilde{\mu }}_{k}\frac{d_{e}x_{k}}{dt}+\sum_{\beta }{\tilde{A}}_{\beta }\;\frac{d\xi_{\beta }}{dt} \end{array} \end{array}$$
where the subscript β counts the existing internal processes. For example,
(33)$$\begin{array}{c} \begin{array}{c} \left\{\frac{d\xi_{\beta }}{dt}\right\}=\left(\begin{array}{c} {\left\{\frac{d\xi_{\alpha }}{dt}\right\}}_{Chem},\;\frac{d\xi_{rv}}{dt},\;\frac{d\xi_{nucl}}{dt},\;\frac{d\xi_{tr}}{dt},\;\\ {\left(\frac{d\xi_{spin}}{dt}\right)}_{\mathrm{NMR}},\;{\left(\frac{d\xi_{spin}}{dt}\right)}_{\mathrm{ESR}},\;\\ \frac{d\xi_{S_{1}S_{0}}}{dt},\;\frac{d\xi_{S_{2}S_{1}}}{dt},\;\frac{d\xi_{S_{1}T_{1}}}{dt},\;\frac{d\xi_{T_{1}S_{0}}}{dt},\;\mathrm{etc}. \end{array}\right) \end{array} \end{array}$$
where the subscripts (i) α refer to various chemical reactions, (ii) rv denotes the rotation–vibration relaxation, (iii) nucl denotes conversion, say, between ortho− and para− hydrogen molecules, (iv) spin denotes involved relaxation between quantum spin states of NMR and ESR spectroscopy, (v) $S_{1}S_{0}\equiv S_{1}\to S_{0}$, $S_{2}S_{1}\equiv S_{2}\to S_{1}$, $S_{1}T_{1}\equiv S_{1}\to T_{1}$, $T_{1}S_{0}\equiv T_{1}\to S_{0}$, etc. The ξ are the extent of advancement of the respective processes.

Broadly speaking (c.f. Figure 6 and Figure 7), except the rates of chemical reactions and the photophysical transition, $T_{1}\to S_{0}$ all are the fast processes. Even in the case of chemical reactions, there are very fast reactions faster than the process of thermalization (even though the reaction can be exo- or endothermic (see for example [63,64])). This is the case of those chemical reactions whose Arrhenius energy of activation, $E_{a}$, is either zero or only a few times of (3/2)RT [65]. That is, one needs to recognize two stages of a reaction. The first one is the actual chemical conversion almost at a constant temperature, whose measured rate constant corresponds to the initial temperature. In the second stage, thermalization takes place, leading to a final temperature higher or lower than the initial temperature (in the case of enthalpy of reaction zero, both the stages of reaction get completed at the constant temperature). Hence, such chemical conversions do fall within the fast domain of time. Therefore, the rates of chemical reactions ${\left\{\frac{d\xi_{\alpha }}{dt}\right\}}_{Chem}$ depending on the fastness of them do fall within the fast domain of time Several excited state chemical reactions also fall well within the fast domain of time (the second order rate constants are of the order $∼10^{10}M^{-1}s^{-1}$ see for example [66]).

Notice that the Gibbs relation of Equation (Equation 32) resembles that of TIV (see for example [30,31,32,33,34,35,36,37,38,39]). However, Equation (Equation 32) is well within the LTE domain, and the physical origin of the involved hidden type of variables is found in the nonequilibrium population of the quantum states.

It is interesting to recall that the Gibbs relation in Hillert’s work in the case of spatially uniform system reads as [7],
(34)$$\begin{array}{c} \begin{array}{c} dU=TdS-pdV+\sum_{k}\mu_{k}dN_{k}-Dd\xi \end{array} \end{array}$$
where $dN_{k}$ is the differential change in the corresponding mole numbers by way of exchange of matter. When we compare the last term Ddξ of Equation (Equation 34) with that in Equation (Equation 32) it is evident that it corresponds to the last term of the latter i.e., $\sum_{\beta }{\tilde{A}}_{\beta }\;\frac{d\xi_{\beta }}{dt}$. It implies that in the proposal of Hillert, all the internal irreversible processes are treated cumulatively. In Hillert’s setting the quantity *D* gets identified as,
(35)$$\begin{array}{c} \begin{array}{c} D\equiv T\frac{d_{i}S}{d\xi }>0 \end{array} \end{array}$$
where *D* is termed as driving force of internal irreversible process (the symbol *D* is used in the honor of De Donder). Notice that the proposal of Hillert conforms with a basic understanding about thermodynamics, that is—thermodynamics is a global description and is immune to the internal mechanism of a given process. Whereas, in Equation (Equation 32) one is guided by the fact that the types of internal irreversible processes identified are distinctly different and belong to a wide range of the time domain and have respectively different global level manifestations as described above. Whereas in the case of chemical reactions their global manifestation is the respective rate of reaction. Recall that by laboratory experimentation one first establishes the functional dependence of rate of reaction on the concentration of various reactants, products, inhibitors, catalysts, etc. Then the reaction steps proposed which are commensurate with the experimentally established expression of rate of reaction. In doing so the Bodenstein steady state approximation for highly reactive intermediate is often applied and if certain pre-equilibrium steps exist they too are accounted for. Moreover, the above description corresponds to the conventional interpretation of LTE. Therefore, it would be of interest to describe corresponding nonequilibrium thermodynamics that obviously would be a description at global level with a better model, close to reality. We foresee that the studies on these lines would enrich the field of CT.

Next we recall that the Gibbs relation of Equation (Equation 9) implies the following functional dependence: (36)$$\begin{array}{c} \begin{array}{c} s=s\left(u,\;v,\;\{{\tilde{x}}_{k,\;j}\}\right) \end{array} \end{array}$$
which gets transformed on using the standard expression of the Gibbs function G=u+pv−Ts to,
(37)$$\begin{array}{c} \begin{array}{c} G=G\left(T,\;p,\;\{{\tilde{x}}_{k,\;j}\}\right) \end{array} \end{array}$$

Therefore, corresponding to the Gibbs relation Equation (Equation 32), the functional dependence gets expressed as,
(38)$$\begin{array}{c} \begin{array}{c} G=G\left(T,\;p,\;\left\{x_{k}\right\},\;\xi_{rv},\;\xi_{nucl},\;\xi_{tr},\;\xi_{spin},\;\xi_{S_{1}S_{0}},\;\xi_{S_{1}T_{1}},\;\mathrm{etc}.\right) \end{array} \end{array}$$
which in the entropy setting reads as
(39)$$\begin{array}{c} \begin{array}{c} s=s\left(u,\;v,\;\left\{x_{k}\right\},\;\xi_{rv},\;\xi_{nucl},\;\xi_{tr},\;\xi_{spin},\;\xi_{S_{1}S_{0}},\;\xi_{S_{1}T_{1}},\;\mathrm{etc}.\right) \end{array} \end{array}$$
where the subscripts (i) rv stands for rotational–vibrational mode, (ii) $\xi_{tr}$ for translational mode, (iii) $\xi_{nucl}$ for the nuclear mode, (iv) $\xi_{spin}$ for nuclear/electron spin modes. The remaining subscripts are described below in Equation (Equation 33).

Corresponding to the expression of Equation (Equation 38), the Gibbs relation reads as,
(40)$$\begin{array}{c} \begin{array}{c} dG=-sdT+vdp+\sum_{k}{\tilde{\mu }}_{k}dx_{k}-{\tilde{A}}_{rv}d\xi_{rv}-{\tilde{A}}_{tr}d\xi_{tr}-\cdots \cdots \end{array} \end{array}$$
and the companion Gibbs function gets expressed as,
(41)$$\begin{array}{c} \begin{array}{c} G=\sum_{k}\;x_{k}\;{\tilde{\mu }}_{k}+\sum_{k,\;j}{\tilde{\mu }}_{rv}^{k,\;j}\;{\tilde{x}}_{rv}^{k,\;j}+\sum_{k,\;j}{\tilde{\mu }}_{tr}^{k,\;j}\;{\tilde{x}}_{tr}^{k,\;j}+\cdots \cdots \end{array} \end{array}$$

Therefore, the equations of state for temperature and pressure read as,
(42)$$\begin{array}{c} \begin{array}{c} \frac{1}{T}={\left(\frac{\partial s}{\partial u}\right)}_{v,\;\left\{x_{k}\right\},\;\left\{\xi_{i}\right\}},\;\frac{p}{T}={\left(\frac{\partial s}{\partial v}\right)}_{u,\;\left\{x_{k}\right\},\;\left\{\xi_{i}\right\}} \end{array} \end{array}$$
where the subscript $\left\{\xi_{i}\right\}$ denote the constancy of all ξ in Equation (Equation 42). However, in view of the vast difference in time scales of the processes, the functional dependence of Equation (Equation 39) gets grouped into two sets.

#### 3.1.1. Gibbs Relation with Internal Variables as Additional Variables: Fast Domain of Time

In the fast domain of time, the functional dependence of *s* reads as,
(43)$$\begin{array}{c} \begin{array}{c} s=s(\left\{x_{k}\right\},\;\xi_{rv},\;\xi_{nucl},\;\xi_{tr},\;\xi_{S_{1}S_{0}},\;\xi_{S_{1}T_{1}},\;\mathrm{etc}.) \end{array} \end{array}$$
provided very fast chemical reactions are taking place. Notice that, though $\left\{x_{k}\right\}$ of Equation (Equation 43) is varying with time, this variation is not due to the exchange of matter because this evolution is under the condition of virtual isolation. Hence, the time variation of *s* under the condition of virtual isolation gets expressed as the virtual constancy of ($u,\;v,\;\{x_{k}\}$). Further, the variation in the *fast domain of time* of $\left\{x_{k}\right\}$ and $\left\{\xi_{tr}\right\}$ proceeds practically at a constant temperature in the virtual isolation condition. Let us illustrate it a little further. $\frac{d\xi_{tr}}{dt}$ represents the internal mechanism of population redistribution operating via the molecular collisional mechanism. Since in the molecular collisions, energy is conserved, the temperature remains practically constant. On the other hand, a chemical reaction may be endo- or exothermic, and the temperature is bound to change. However, when chemical reactions are very fast they do fall in the fast domain of time, that is the chemical conversion gets completed before thermalization is initiated. Hence, in the first fast time domain, the temperature remains practically constant. The cases of fast reactions with zero enthalpy of reaction ${\left(\frac{\partial H}{\partial \xi }\right)}_{T,\;p}=0$ in which the free energy of reaction gets represented by ${\left(\frac{\partial G}{\partial \xi }\right)}_{T,\;p}=-T{\left(\frac{\partial s}{\partial \xi }\right)}_{T,\;p}<0$, which is entirely entropy production due to the balance between mixing and demixing processes (see for example [67]), purely a non-thermal process comparable to the mixing of two ideal gases initially at identical *T* and *p*. The examples of such reactions are the electron exchange reactions between transition metal ions that follow the outer sphere mechanism (for example $\mathrm{Cr}{{\left(\mathrm{bipy}\right)}_{3}}^{1+}+\mathrm{Cr}{\left(\mathrm{bipy}\right)}_{3}=\mathrm{Cr}{\left(\mathrm{bipy}\right)}_{3}+\mathrm{Cr}{{\left(\mathrm{bipy}\right)}_{3}}^{1+}$. See, for example, [66,68,69]) (k = (1.5 $\pm \;0.4)\times 10^{9}M^{-1}s^{-1}$).

On the other hand, in the *slow domain of time*, the operating functional dependence reads as,
(44)$$\begin{array}{c} \begin{array}{c} s=s(u,\;v,\;\left\{x_{k}\right\},\;\xi_{spin},\;\xi_{T_{1}S_{0}},\;\mathrm{etc}.) \end{array} \end{array}$$

Herein, the variation of $\left\{x_{k}\right\}$ includes relatively slower chemical reactions. The other processes included are the spin relaxation process and the triplate to singlet transformation, which is a well-known slow process, whose variable is $\xi_{T_{1}S_{0}}$. It also implies the operation of the functional dependencies $\xi_{rv}=\xi_{rv}\left(u,\;v,\;\left\{x_{k}\right\},\;\xi_{T_{1}S_{0}}\right),\;\;$$\xi_{tr}=\xi_{tr}\left(u,\;v,\;\left\{x_{k}\right\},\;\xi_{T_{1}S_{0}}\right),\;\;\;$$\xi_{S_{1}T_{1}}=\xi_{S_{1}T_{1}}\left(u,\;v,\;\left\{x_{k}\right\},\;\xi_{T_{1}S_{0}}\right),\;\;\;$$\xi_{nucl}=\xi_{nucl}\left(u,\;v,\;\left\{x_{k}\right\},\;\xi_{T_{1}S_{0}}\right),\;$ etc.

Therefore, in the view of the equations of state of Equation (Equation 42), the equations of state corresponding to the functional dependence of Equation (Equation 43) read as
(45)$$\begin{array}{c} \begin{array}{c} \begin{array}{c} \frac{{\tilde{A}}_{\alpha }}{T}={\left(\frac{\partial s}{\partial \xi_{\alpha }}\right)}_{u,\;v,\;\{d_{e}x_{k}=0\},\;\xi }=-\frac{1}{T}\sum_{k}{\tilde{\mu }}_{k}M_{k}\nu_{k}^{\alpha },\;\frac{{\tilde{A}}_{rv}}{T}={\left(\frac{\partial s}{\partial \xi_{rv}}\right)}_{u,\;v,\;x,\;\xi^{\prime }\neq \xi_{rv}},\\ \frac{{\tilde{A}}_{tr}}{T}={\left(\frac{\partial s}{\partial \xi_{tr}}\right)}_{u,\;v,\;x,\;\xi^{\prime }\neq \xi_{tr}},\;\mathrm{etc}.\;\;\; \end{array} \end{array} \end{array}$$
where the subscript ξ in the first equation of state of Equation (Equation 45) stands for the constancy of all ξ, including all extents of the advancement of chemical reactions $\xi_{\beta }$, except $\xi_{\beta =\alpha }$ and $\left\{d_{e}x_{k}=0\right\}$ which need to be specified to ensure the non-contribution due to the matter exchange.

The expression of ${\tilde{A}}_{rv}$ reads as,
(46)$$\begin{array}{c} \begin{array}{c} {\tilde{A}}_{rv}=\sum_{k}\left(\mu_{k,\;rv}-{\tilde{\mu }}_{k,\;rv}\right)M_{k} \end{array} \end{array}$$
with
(47)$$\begin{array}{c} \begin{array}{c} {\tilde{\mu }}_{k,\;rv}=\sum_{j}{\tilde{\mu }}_{rv}^{k,\;j}\;{\tilde{\gamma }}_{rv}^{{}^{k,\;j}},\;\mu_{k,\;rv}=\sum_{j}\gamma_{rv}^{{}^{k,\;j}}\mu_{rv}^{k,\;j}=\sum_{j}\gamma_{rv}^{{}^{k,\;j}}\mu_{k,\;rv}=\mu_{k,\;rv} \end{array} \end{array}$$
where $\mu_{k,\;rv}$ refers to the population equilibrated rotational–vibrational quantum levels. In this state, we have $\mu_{rv}^{k,\;j}=\mu_{rv}^{k,\;j^{\prime }}=\mu_{rv}^{k,\;j^{\prime \prime }}=\cdots =\mu_{k,\;rv}$ and $\sum_{j}\gamma_{rv}^{{}^{k,\;j}}=1$. A similar expression for ${\tilde{A}}_{tr}$ can be written down.

Thus, notice that though the direct equation of state for temperature *T* cannot be written using the the functional dependency of Equation (Equation 43) but indirectly, we are led to the function *T* in the equations of state of Equation (Equation 45), this is because *T* remains virtually constant and also because no heat gets exchanged at the local pocket in this *fast domain of time*.

The functional dependency of Equation (Equation 44), that is, for the *slow domain of time*, does gives direct equation of state for the temperature function, *T*, that reads as
(48)$$\begin{array}{c} \begin{array}{c} \frac{1}{T}={\left(\frac{\partial s}{\partial u}\right)}_{v,\;x,\;\xi_{T_{1}S_{0}}} \end{array} \end{array}$$
where we assumed that, except $\xi_{T_{1}S_{0}}$, no additional slow variable exists.

Thus in the *fast domain of time* the Gibbs relation reads as,
(49)$$\begin{array}{c} \begin{array}{c} \begin{array}{cc} \rho \frac{ds}{dt}=\sigma_{s}= & \sum_{\alpha }\frac{{\tilde{A}}_{\alpha }}{T}\frac{d\xi_{\alpha }}{dt}+\frac{{\tilde{A}}_{rv}}{T}\frac{d\xi_{\alpha }}{dt}+\frac{{\tilde{A}}_{nucl}}{T}\frac{d\xi_{nucl}}{dt}+\frac{{\tilde{A}}_{tr}}{T}\frac{d\xi_{tr}}{dt}\\ \; & +{\left(\frac{{\tilde{A}}_{spin}}{T}\right)}_{\mathrm{NMR}}{\left(\frac{d\xi_{spin}}{dt}\right)}_{\mathrm{NMR}}+{\left(\frac{{\tilde{A}}_{spin}}{T}\right)}_{\mathrm{ESR}}{\left(\frac{d\xi_{spin}}{dt}\right)}_{\mathrm{ESR}}\\ \; & +\frac{{\tilde{A}}_{\xi_{S_{1}S_{0}}}}{T}\frac{d\xi_{S_{1}S_{0}}}{dt}+\frac{{\tilde{A}}_{\xi_{S_{1}T_{1}}}}{T}\frac{d\xi_{S_{1}T_{1}}}{dt}+\mathrm{etc}.>0 \end{array} \end{array} \end{array}$$

It is entirely the rate of entropy production because the evolution is under the condition of virtual isolation.

#### 3.1.2. Gibbs Relation with Internal Variables as Additional Variables: Slow Domain of Time

In the *slow domain of time* the Gibbs relation reads as,
(50)$$\begin{array}{c} \begin{array}{c} T\frac{ds}{dt}=\frac{du}{dt}+p\frac{dv}{dt}-\sum_{k}{\tilde{\mu }}_{k}\frac{d_{e}x_{k}}{dt}+\sum_{\alpha }{\tilde{A}}_{\alpha }\;\frac{d\xi_{\alpha }}{dt}+{\tilde{A}}_{\xi_{T_{1}S_{0}}}\frac{d\xi_{T_{1}S_{0}}}{dt}+\mathrm{etc}. \end{array} \end{array}$$
which on using the internal energy and mass balance equations of Equations (Equation 6) and (Equation 7), the expressions of entropy source strength and entropy flux density read as
(51)$$\begin{array}{c} \begin{array}{c} \begin{array}{cc} \sigma_{s}= & q\cdot \nabla \left(\frac{1}{T}\right)+\frac{1}{T}\Pi :\nabla u-\frac{1}{T}\sum_{k}J_{k}\cdot \left(T\nabla \left(\frac{{\tilde{\mu }}_{k}}{T}\right)-F_{k}\right)\\ \; & +\sum_{\alpha }\frac{{\tilde{A}}_{\alpha }}{T}\frac{d\xi_{\alpha }}{dt}+\frac{{\tilde{A}}_{\xi_{T_{1}S_{0}}}}{T}\frac{d\xi_{T_{1}S_{0}}}{dt}+\mathrm{etc}.>0 \end{array} \end{array} \end{array}$$
and
(52)$$\begin{array}{c} \begin{array}{c} J_{s}=\frac{q-\sum_{k}{\tilde{\mu }}_{k}J_{k}}{T}≷0 \end{array} \end{array}$$

Compare the expressions of Equations (Equation 26) and (Equation 51). This difference is due to the difference between the Gibbs relations Equations (Equation 25) and (Equation 50), respectively.

### 3.2. Gibbs Relation of GPITT with Physical Fluxes as Additional Variables

The existence of physical fluxes q, Π and $J_{k}$ that correspond to the nonequilibrium population of the translational quantum states allows us to extract out this contribution from the variables ${\tilde{x}}_{k,\;j}$ and replace them by the said physical fluxes. In other words, Equation (Equation 33) is expressed as
(53)$$\begin{array}{c} \begin{array}{c} \left\{\frac{d\xi_{\beta }}{dt}\right\}=\left(\begin{array}{c} {\left\{\frac{d\xi_{\alpha }}{dt}\right\}}_{Chem},\;\frac{d\xi_{rv}}{dt},\;\frac{d\xi_{nucl}}{dt},\;\frac{dq}{dt},\;\frac{d\Pi }{dt},\;\left\{\frac{dJ_{k}}{dt}\right\},\;\\ {\left(\frac{d\xi_{spin}}{dt}\right)}_{\mathrm{NMR}},\;{\left(\frac{d\xi_{spin}}{dt}\right)}_{\mathrm{ESR}},\;\\ \frac{d\xi_{S_{1}S_{0}}}{dt},\;\frac{d\xi_{S_{2}S_{1}}}{dt},\;\frac{d\xi_{S_{1}T_{1}}}{dt},\;\frac{d\xi_{T_{1}S_{0}}}{dt},\;\mathrm{etc}. \end{array}\right) \end{array} \end{array}$$

For the sake of simplicity, we assume that only sources of internal irreversibility are due to chemical reactions, the existence of fluxes and the nonequilibrium population of rotational–vibrational quantum states. Thus, in view of the above facts the functional dependence of Equations (Equation 37) and (Equation 38) is represented as
(54)$$\begin{array}{c} \begin{array}{c} G=G\left(T,\;p,\;\{{\tilde{x}}_{k,\;j}\}\right)\equiv G\left(T,\;p,\;\left\{x_{k}\right\},\;\xi_{rv},\;q,\;\Pi ,\;\left\{J_{k}\right\}\right) \end{array} \end{array}$$

For the sake of simplicity, we consider the systems having nonequilibrium population only of translation, rotation, and vibration quantum levels. Correspondingly, the Gibbs relation of GPITT reads as
(55)$$\begin{array}{c} \begin{array}{c} dG=-sdT+vdp+\sum_{k}{\tilde{\mu }}_{k}dx_{k}-{\tilde{A}}_{rv}d\xi_{rv}+\beta_{q}q\cdot dq+\beta_{\Pi }\Pi :d\Pi +\sum_{k}\beta_{J_{k}}J_{k}\cdot dJ_{k} \end{array} \end{array}$$

In this way, the Gibbs function, G, is expressed as
(56)$$\begin{array}{c} \begin{array}{c} G=\sum_{k}\;x_{k}\;{\tilde{\mu }}_{k}+\sum_{k,\;j}{\tilde{\mu }}_{rv}^{k,\;j}\;{\tilde{x}}_{rv}^{k,\;j}+\frac{1}{2}\beta_{q}q^{2}+\frac{1}{2}\beta_{\Pi }\Pi^{2}+\frac{1}{2}\sum_{k}\beta_{J_{k}}J_{k}^{2} \end{array} \end{array}$$
where the subscript rv refers to the rotational–vibrational degrees of freedom, $\beta_{q}$, $\beta_{\Pi }$ and $\beta_{J_{k}}$ are the corresponding physical coefficients whose physical meaning yet to be established but need to be corresponding intensive quantities. Thus, the five terms on the r.h.s. of Equation (Equation 56) are the individual contribution of irreversibility accounted by the single term $\sum_{k,\;j}{\tilde{\mu }}_{k,\;j}{\tilde{x}}_{k,\;x}$ of Equation (Equation 22). It is somewhat similar to the Born–Oppenheimer approximation that leads us to consider that the energies from each type of motion are additive. That is, for example, we use the splitting $E_{total}=E_{electronic}+E_{vibrational}+E_{rotational}$ on account of separate contributions from the electronic, vibrational and rotational molecular motions.

The equations of state corresponding to the Gibbs relations Equation (Equation 55) read as
(57)$$\begin{array}{c} \begin{array}{c} -s={\left(\frac{\partial G}{\partial T}\right)}_{p,\;{x_{k}}^{\prime }s,\;q,\;\Pi ,\;{J_{k}}^{\prime }s,\;\xi_{rv}} \end{array} \end{array}$$
(58)$$\begin{array}{c} \begin{array}{c} v={\left(\frac{\partial G}{\partial p}\right)}_{T,\;{x_{k}}^{\prime }s,\;q,\;\Pi ,\;{J_{k}}^{\prime }s,\;\xi_{rv}} \end{array} \end{array}$$
(59)$$\begin{array}{c} \begin{array}{c} {\tilde{\mu }}_{k}={\left(\frac{\partial G}{\partial x_{k}}\right)}_{T,\;p,\;x^{\prime },\;q,\;\Pi ,\;{J_{k}}^{\prime }s,\;\xi_{rv}} \end{array} \end{array}$$
(60)$$\begin{array}{c} \begin{array}{c} {\tilde{A}}_{rv}=-{\left(\frac{\partial G}{\partial \xi_{rv}}\right)}_{T,\;p,\;{x_{k}}^{\prime }s,\;q,\;\Pi ,\;{J_{k}}^{\prime }s} \end{array} \end{array}$$
and we adopt the linear relations for the remaining equations of state,
(61)$$\begin{array}{c} \begin{array}{c} \beta_{q}q={\left(\frac{\partial G}{\partial q}\right)}_{T,\;p,\;{x_{k}}^{\prime }s,\;\Pi ,\;{J_{k}}^{\prime }s,\;\xi_{rv}} \end{array} \end{array}$$
(62)$$\begin{array}{c} \begin{array}{c} \beta_{\Pi }\Pi ={\left(\frac{\partial G}{\partial \Pi }\right)}_{T,\;p,\;{x_{k}}^{\prime }s,\;q,\;{J_{k}}^{\prime }s,\;\xi_{rv}} \end{array} \end{array}$$
(63)$$\begin{array}{c} \begin{array}{c} \beta_{J_{k}}J_{k}={\left(\frac{\partial G}{\partial J_{k}}\right)}_{T,\;p,\;{x_{k}}^{\prime }s,\;\Pi ,\;q,\;J^{\prime },\;\xi_{rv}} \end{array} \end{array}$$

The expression of ${\tilde{A}}_{rv}$ of Equation (Equation 60) is the same that given in Equation (Equation 46). Later on, we will see that the equations of state of Equations (Equation 57) to (Equation 60) belong to the *slow domain of time*. The physical contents of the coefficients $\beta_{q}q$, $\beta_{\Pi }\Pi$ and $\beta_{J_{k}}J_{k}$ need to be established.

The Gibbs relation of Equation (Equation 55) in the entropy settings reads as,
(64)$$\begin{array}{c} \begin{array}{c} T\frac{ds}{dt}\;=\;\frac{du}{dt}\;+\;p\frac{dv}{dt}\;-\sum_{k}{\tilde{\mu }}_{k}\frac{dx_{k}}{dt}+\frac{{\tilde{A}}_{rv}}{\rho }\frac{d\xi_{rv}}{dt}-\beta_{q}q\cdot \frac{dq}{dt}-\beta_{\Pi }\Pi :\frac{d\Pi }{dt}-\sum_{k}\beta_{J_{k}}J_{k}\cdot \frac{dJ_{k}}{dt} \end{array} \end{array}$$
and the accompanying Gibbs–Duhem equation is
(65)$$\begin{array}{c} \begin{array}{c} \sum_{k}x_{k}\;\frac{d{\tilde{\mu }}_{k}}{dt}+s\frac{dT}{dt}-v\frac{dp}{dt}+\sum_{k,\;j}{\tilde{x}}_{rv}^{k,\;j}\frac{d{\tilde{\mu }}_{rv}^{k,\;j}}{dt}+\frac{1}{2}q^{2}\frac{d\beta_{q}}{dt}+\frac{1}{2}\Pi^{2}\frac{d\beta_{\Pi }}{dt}+\frac{1}{2}\sum_{k}J_{k}^{2}\frac{d\beta_{J_{k}}}{dt}=0 \end{array} \end{array}$$
where we used the following relations for the mass balance and the chemical affinity of the rotational–vibrational equilibration process:(66)$$\begin{array}{c} \begin{array}{c} \rho \frac{d{\tilde{x}}_{rv}^{k,\;j}}{dt}=\nu_{rv}^{k,\;j}M_{k}\frac{d\xi_{rv}}{dt},\;{\tilde{A}}_{rv}=-\sum_{k,\;j}\nu_{rv}^{k,\;j}{\tilde{\mu }}_{rv}^{k,\;j}{\tilde{\gamma }}_{rv}^{k,\;j}M_{k} \end{array} \end{array}$$

The expression of rotational–vibrational affinity in Equation (Equation 66) produces its expression contained in Equation (Equation 46) by recalling that the stoichiometric coefficients $\nu_{rv}^{k,\;j}$ have either +1 or −1 value. Additionally, notice that this EIT-type Gibbs relation Equation (Equation 64) is indeed one of the version of the TIV-type Gibbs relation, say, of Section 3.1.

Thus, it is apparent that the entire development of GPITT in the present version resembles that of EIT with the difference that in the latter, they use the so-called nonequilibrium quantities θ and π (refer, for example, to [23]). In contrast, the GPITT is well within the LTE domain. In other words, the GPITT removes the incompleteness that exists in the conventional CIT (though it has not been realized so far). Thus, it is a clear and direct demonstration that incorporating the physical fluxes and other macroscopic variables, such as those of Equation (Equation 53), does not lead to the breakdown of LTE. A similar type of conclusion was arrived at earlier about the quality of temperature and pressure in EIT [70].

The next subject matter that we investigated is the use of various already existing constitutive equations for the physical fluxes. For the sake of convenience in the next Section 3.2.1, we list constitutive equations that we use in this presentation.

#### 3.2.1. Constitutive Equations of Physical Fluxes for a Spatially Non-Uniform System

Let us recall a few of the constitutive equations. For heat conduction, (i) the Maxwell–Cattaneo equation [71,72]: (67)$$\begin{array}{c} \begin{array}{c} \tau_{q}\frac{dq}{dt}+q=-\lambda \nabla T, \end{array} \end{array}$$
where $\tau_{q}$ is the relaxation time for the decay of the heat flux density and λ is the thermal conductivity, (ii) the Guyer–Krumhansl’s relation, [72,73,74],
(68)$$\begin{array}{c} \begin{array}{c} \tau_{q}\frac{dq}{dt}+q=-\lambda \nabla T+l^{2}\nabla^{2}q+2l^{2}\nabla \left(\nabla \cdot q\right) \end{array} \end{array}$$
where *l* is the mean free path of phonons, and (iii) Jeffreys’-type equation [75],
(69)$$\begin{array}{c} \begin{array}{c} \tau_{q}\frac{dq}{dt}+q=-\lambda \nabla T-\tau_{q}k\frac{d\nabla T}{dt} \end{array} \end{array}$$
k is the effective thermal conductivity.

For the dissipative stress tensor, (i) the Maxwell–Cattaneo type (it is simply known as Maxwell’s model),
(70)$$\begin{array}{c} \begin{array}{c} \tau_{\Pi }\frac{d\Pi }{dt}+\Pi =2\eta \nabla u, \end{array} \end{array}$$
where η is the shear viscosity, and $\tau_{\Pi }$ is the relaxation time of the decay of the dissipative momentum flux density, (ii) Giesekus upper convected model [75],
(71)$$\begin{array}{c} \begin{array}{c} \tau_{\Pi }\frac{d\Pi }{dt}+\Pi +\frac{\alpha }{G}\Pi \cdot \Pi =2\eta \nabla u, \end{array} \end{array}$$
where α is the dimensionless mobility factor (also referred to as the Giesekus parameter), and $G=\eta /\tau_{\Pi }$ is the elasticity modulus of material, (iii) Leonov model,
(72)$$\begin{array}{c} \begin{array}{c} \tau_{\Pi }\frac{d\Pi }{dt}+\Pi +\frac{1}{2G}\Pi \cdot \Pi =2\eta \nabla u. \end{array} \end{array}$$
(iv) Jeffreys type [76,77],
(73)$$\begin{array}{c} \begin{array}{c} \tau_{\Pi }\frac{d\Pi }{dt}+\Pi =2\eta \nabla u+2\eta \;\tau_{\Pi }^{\prime }\frac{d\nabla u}{dt} \end{array} \end{array}$$
where $\tau_{\Pi }$ is the relaxation time for the decay of the dissipative momentum flux density, $\tau_{\Pi }^{\prime }$ is the retardation time of the viscoelastic fluid, and for diffusion flux (i) Maxwell–Cattaneo type,
(74)$$\begin{array}{c} \begin{array}{c} \tau_{J_{k}}\frac{dJ_{k}}{dt}+J_{k}=-\rho D_{k}\nabla c_{k} \end{array} \end{array}$$
where $\tau_{J_{k}}$ is the relaxation time of the decay of matter diffusion flux density, $D_{k}$ is the diffusion coefficient of the component *k* and (ii) Jeffreys type [78],
(75)$$\begin{array}{c} \begin{array}{c} \tau_{J_{k}}\frac{dJ_{k}}{dt}+J_{k}=-D_{k}\rho \left(\tau_{J_{k}}^{\prime }\frac{d\nabla c_{k}}{dt}+\nabla c_{k}\right) \end{array} \end{array}$$
where $\tau_{J_{k}}^{\prime }$ is the fractionally weighted relaxation time for the time rate of change of the concentration gradient. In the above expressions, τ shows the respective relaxation times.

#### 3.2.2. Gibbs Relation with Physical Fluxes as Additional Variables. Implications of Fast and Slow Variables

As stated above, the whole evolution proceeds in two time domains. In the *fast time domain*, the Gibbs relation of Equation (Equation 64) (on ignoring the term quantifying rotation-vibration equilibration process) reads effectively as,
(76)$$\begin{array}{c} \begin{array}{c} T\frac{ds}{dt}\;=-\beta_{q}q\cdot \frac{dq}{dt}-\beta_{\Pi }\Pi :\frac{d\Pi }{dt}-\sum_{k}\beta_{J_{k}}J_{k}\cdot \frac{dJ_{k}}{dt} \end{array} \end{array}$$
as the variables ($u,\;v,\;\{x_{k}\}$) remain practically constant. In the *slow domain of time*, the Gibbs relation of Equation (Equation 64) reads effectively as Equation (Equation 25) and the Gibbs–Duhem equation of Equation (Equation 65) reads as the expression of Equation (Equation 28).

However, notice that the Gibbs relation of Equation (Equation 25) in view of the above discussion gets specified as a description in the *slow domain of time* and by default is accompanied with the functional dependencies $q=q(u,\;v,\;\left\{x_{k}\right\})$, $\Pi =\Pi (u,\;v,\;\left\{x_{k}\right\})$ and for all $J_{k}=J_{k}\left(u,\;v,\;\left\{x_{l}\right\}\right)$. This fact does not get impressed upon if we use ${\tilde{\mu }}_{k}$ directly in place of $\mu_{k}$ as is described below in Equation (Equation 23) in Section 2. However, let us check the equations of state of those operating in the *slow domain of time* based on Equation (Equation 25): (77)$$\begin{array}{c} \begin{array}{c} {\left(\frac{\partial s}{\partial u}\right)}_{v,\;x}=\frac{1}{T}\;,\;{\left(\frac{\partial s}{\partial v}\right)}_{u,\;x}=\frac{p}{T}\;,\;{\left(\frac{\partial s}{\partial x_{k}}\right)}_{u,\;v,\;x^{\prime }}=\frac{{\tilde{\mu }}_{k}}{T} \end{array} \end{array}$$
where the subscript *x* means all mass fractions are kept constant and $x^{\prime }$ denotes keeping all mass fractions constant, except $x_{k}$. Notice that there is no need to specify the constancy of physical fluxes herein because they no more remain independent thermodynamic variables, but they do exist in the system during the evolution in the *slow domain of time*. The situation in the *fast domain of time* is different because there, the physical fluxes play the role of independent variables and hence, for writing down equations of state from Equation (Equation 76), by default, the constancy of $\left(u,\;v,\;\left\{x_{l}\right\}\right)$ gets prescribed. Therefore, the operative Gibbs relation of the *fast domain of time*, Equation (Equation 76), does not offer a direct equation of state for temperature *T* but the indirect or implied ones appear in the following equations of state: (78)$$\begin{array}{c} \begin{array}{c} \begin{array}{c} {\left(\frac{\partial s}{\partial q}\right)}_{u,\;v,\;x,\;\Pi ,\;J_{k}^{\prime }s}=-\frac{\beta_{q}q}{T}\;,\;{\left(\frac{\partial s}{\partial \Pi }\right)}_{u,\;v,\;x,\;q,\;J_{k}^{\prime }s}=-\frac{\beta_{\Pi }\Pi }{T}\;,\\ {\left(\frac{\partial s}{\partial J_{k}}\right)}_{u,\;v,\;x,\;\Pi ,\;J^{\prime }s}=-\frac{\beta_{J_{k}}J_{k}}{T}\;\;\; \end{array} \end{array} \end{array}$$

The constancy of $u,\;v,\;\{x_{k}\}$ used in the preceding definitions stems from the fact that the Gibbs relation of Equation (Equation 76) operates under the virtual condition of constancy of $u,\;v,\;\{x_{k}\}$ or in other words, under the condition of virtual isolation.

Geometrically speaking, let us consider a thermodynamic multi-dimensional space
(79)$$\begin{array}{c} \begin{array}{c} \left(s,\;u,\;v,\;\left\{x_{k}\right\},\;q,\;\Pi ,\;\left\{J_{k}\right\}\right) \end{array} \end{array}$$
provided by the Gibbs relation Equation (Equation 64) (for the sake of simplicity, we have ignored the variable $\xi_{rv}$) whose solution hyper-surfaces would read as
(80)$$\begin{array}{c} \begin{array}{c} s=s(u,\;v,\;\left\{x_{k}\right\},\;q,\;\Pi ,\;\left\{J_{k}\right\})\;=\mathrm{constant} \end{array} \end{array}$$

The projection of the multi-dimensional space of Equation (Equation 79) under the condition of constancy of ($u,\;v,\;\{x_{k}\}$) gives the reduced multi-dimensional space
(81)$$\begin{array}{c} \begin{array}{c} \left(s,\;q,\;\Pi ,\;\left\{J_{k}\right\}\right)\;\mathrm{with}(u,\;v,\;\left\{x_{k}\right\})=\mathrm{constant} \end{array} \end{array}$$
whose solution hyper-surface reads as
(82)$$\begin{array}{c} \begin{array}{c} s=s\left(q,\;\Pi ,\;\left\{J_{k}\right\}\right)\;=\mathrm{constant};\mathrm{with}(u,\;v,\;\left\{x_{k}\right\})=\mathrm{constant} \end{array} \end{array}$$
which corresponds to the Gibbs relation of Equation (Equation 76) of the *fast domain of time*.

On the other hand, under the conditions of $q=q(u,\;v,\;\left\{x_{k}\right\})$, $\Pi =\Pi (u,\;v,\;\left\{x_{k}\right\})$ and for all $J_{k}=J_{k}\left(u,\;v,\;\left\{x_{l}\right\}\right)$, the respective projections of Equations (Equation 79) and (Equation 80) read as,
(83)$$\begin{array}{c} \begin{array}{c} \begin{array}{c} (s,\;u,\;v,\;\left\{x_{k}\right\})\;\;\;\;\;\;\\ \mathrm{with}q=q\left(u,\;v,\;\left\{x_{k}\right\}\right),\;\Pi =\Pi \left(u,\;v,\;\left\{x_{k}\right\}\right),\;\mathrm{for}\;\mathrm{all}\;J_{k}=J_{k}\left(u,\;v,\;\left\{x_{l}\right\}\right) \end{array} \end{array} \end{array}$$
and
(84)$$\begin{array}{c} \begin{array}{c} \begin{array}{c} s=s(\;u,\;v,\;\left\{x_{k}\right\})=\mathrm{constant}\;\;\;\;\;\\ \mathrm{with}q=q\left(u,\;v,\;\left\{x_{k}\right\}\right),\;\;\Pi =\Pi \left(u,\;v,\;\left\{x_{k}\right\}\right),\;\;\mathrm{for}\;\mathrm{all}\;J_{k}=J_{k}\left(u,\;v,\;\left\{x_{l}\right\}\right) \end{array} \end{array} \end{array}$$

The above geometrical description lucidly illustrates the correspondences of the equations of state in *fast and slow domains of time.* Notice also that the origin of the functional dependencies of physical fluxes shown in Equations (Equation 83) and (Equation 84) get clearly established. Hence, the correct description in the presence of physical fluxes and in the *slow domain of time* is Equation (Equation 25) accompanied with the functional dependencies shown in Equation (Equation 84) but not that of Equation (Equation 2). Moreover, the thermodynamic space described by the expression of Equation (Equation 79) is not the one in which the system evolves but the real thermodynamic spaces are the ones described by Equations (Equation 81) and (Equation 83); the former corresponds to the *fast domain of time* and the latter one is that for the *slow domain of time*.

Recall that the above versions of Gibbs relations Equations (Equation 9), (Equation 25), (Equation 29), (Equation 30), (Equation 32), (Equation 40), (Equation 50), (Equation 55), (Equation 64) and (Equation 76) fall well within the LTE domain. However, in the *fast domain of time*, the variables $\left(u,\;v,\;\left\{x_{k}\right\}\right)$ remain practically constant, amounting to the absence of exchange processes; hence, the Gibbs relation of Equation (Equation 76) is effectively expressed as
(85)$$\begin{array}{c} \begin{array}{c} \rho \frac{ds}{dt}\;=\;\sigma_{s}=-\frac{\rho }{T}\beta_{q}q\cdot \frac{dq}{dt}-\frac{\rho }{T}\beta_{\Pi }\Pi :\frac{d\Pi }{dt}-\frac{\rho }{T}\sum_{k}\beta_{J_{k}}J_{k}\cdot \frac{dJ_{k}}{dt}>0 \end{array} \end{array}$$
and it too belongs to the LTE domain and the positive sign of $\sigma_{s}$ of Equation (Equation 85) straightforwardly follows because this stage of evolution is virtually in isolation. It would be profitable to analyze the demonstration by Balescu [79] about the entropy source strength based on the Boltzmann integro-differential equation and Boltzmann H-function [55,57] for various time zones (also discussed in [80]) under the condition operative in the *fast domain of time*. However, these details, in the present context, will be discussed separately at a later date.

At the same time, in the *fast domain of time*, the mass balance rule of Equation (Equation 11) would read as,
(86)$$\begin{array}{c} \begin{array}{c} \rho \;\frac{d_{i}{\tilde{x}}_{k,\;j}}{dt}\;=\nu_{k,\;j}\;M_{k}\frac{d\omega }{dt}\;+\;\sum_{\alpha }\nu_{k}^{\alpha }M_{k}{\tilde{\gamma }}_{{}_{k,\;j}}\;\frac{d\xi_{\alpha }}{dt}, \end{array} \end{array}$$

This is because in the *fast domain of time* none of the $x_{k}$ vary with time. Moreover, the use of Equation (Equation 79) implies $\nabla \cdot {\tilde{J}}_{k,\;j}=0$, but it does not imply $J_{k}=0$ and ${\tilde{J}}_{k,\;j}=0$; hence, in Equation (Equation 76) and in Equation (Equation 85), there we have the last summational term on their r.h.s.

#### 3.2.3. Rate of Change of Entropy in the Fast Domain of Time Using Constitutive Equations for the Physical Fluxes

The physical processes in the *fast domain of time* are those which get initiated as soon as the respective gradients are imposed or get imposed upon and till the fluxes lose their status as independent thermodynamic variables.

We recall that since the physical fluxes are the fast variables, all the constitutive equations of Section 3.2.1 may be used in the *fast domain of time* but with the implied constraint of constancy of ($u,\;v,\;\{x_{k}\}$).

The implied constraints in the *fast domain of time* are that (i) all the divergence terms practically remain equal to zero and (ii) all the gradient terms remain constant. As a result of this, Equations (Equation 67)–(Equation 69) would practically read as,
(87)$$\begin{array}{c} \begin{array}{c} \tau_{q}\frac{dq}{dt}+q=-\lambda \nabla T=\mathrm{constant} \end{array} \end{array}$$

Equations (Equation 70) and (Equation 73) would practically read as,
(88)$$\begin{array}{c} \begin{array}{c} \tau_{\Pi }\frac{d\Pi }{dt}+\Pi =2\eta \nabla u=\mathrm{constant} \end{array} \end{array}$$
and Equations (Equation 74) and (Equation 75) would practically read as,
(89)$$\begin{array}{c} \begin{array}{c} \tau_{J_{k}}\frac{dJ_{k}}{dt}+J_{k}=-\rho D_{k}\nabla c_{k}=\mathrm{constant} \end{array} \end{array}$$

Notice that all the above expressions are of the Maxwell–Cattaneo type, whereas the constitutive equations for visco-elastic fluids, that is, Equations (Equation 71) and (Equation 72) in the *fast domain of time,* read as,
(90)$$\begin{array}{c} \begin{array}{c} \tau_{\Pi }\frac{d\Pi }{dt}+\Pi +\frac{\alpha }{G}\Pi \cdot \Pi =2\eta \nabla u=\mathrm{constant} \end{array} \end{array}$$
and
(91)$$\begin{array}{c} \begin{array}{c} \tau_{\Pi }\frac{d\Pi }{dt}+\Pi +\frac{1}{2G}\Pi \cdot \Pi =2\eta \nabla u=\mathrm{constant} \end{array} \end{array}$$

Therefore, in the *fast domain of time* we have the following description of entropy source strength wherein we used Equation (Equation 85) and the preceding five constitutive equations.

In the case of *Maxwell–Cattaneo-type* constitutive equations given in Equations (Equation 87) to (Equation 89), the entropy source strength for the *fast domain of time* reads as,
(92)$$\begin{array}{c} \begin{array}{c} \begin{array}{cc} \sigma_{s}= & \frac{\rho \beta_{q}}{T\tau_{q}}q^{2}+\frac{\rho \beta_{\Pi }}{T\tau_{\Pi }}\Pi^{2}+\frac{\rho }{T}\sum_{k}\frac{\beta_{J_{k}}}{\tau_{J_{k}}}J_{k}^{2}+\frac{\rho \beta_{q}\lambda }{T\tau_{q}}q\cdot \left(\nabla T\right){}_{const}\\ \; & -\frac{2\rho \eta \beta_{\Pi }}{T\tau_{\Pi }}\Pi :\left(\nabla u\right){}_{const}+\frac{\rho^{2}}{T}\sum_{k}\frac{\beta_{J_{k}}D_{k}}{\tau_{J_{k}}}J_{k}\cdot \left(\nabla c_{k}\right){}_{const} \end{array} \end{array} \end{array}$$It is easy to realize that the last three terms on the r.h.s. of Equation (Equation 92) are individually negative, provided that the β-coefficients are positive quantities. However, in this event, the second law of thermodynamics guarantees that the magnitude of the sum of the squared terms preceding them remains greater than the sum of the gradients involved terms. Additionally, with the variation of time, the last three terms on the r.h.s. of Equation (Equation 92) contribute only via the variation of respective fluxes.In the case of visco-elastic fluids using Equations (Equation 90) and (Equation 91) the entropy source strength reads as,
(93)$$\begin{array}{c} \begin{array}{c} \sigma_{s}=\frac{\rho \beta_{\Pi }}{T\tau_{\Pi }}\Pi^{2}-\frac{2\rho \eta \beta_{\Pi }}{T\tau_{\Pi }}\Pi :\left(\nabla u\right){}_{const}+\frac{\rho \alpha \beta_{\Pi }}{T\tau_{\Pi }G}\Pi :\left(\Pi \cdot \Pi \right) \end{array} \end{array}$$
and
(94)$$\begin{array}{c} \begin{array}{c} \sigma_{s}=\frac{\rho \beta_{\Pi }}{T\tau_{\Pi }}\Pi^{2}-\frac{2\rho \eta \beta_{\Pi }}{T\tau_{\Pi }}\Pi :\left(\nabla u\right){}_{const}+\frac{\rho \beta_{\Pi }}{2T\tau_{\Pi }G}\Pi :\left(\Pi \cdot \Pi \right) \end{array} \end{array}$$
respectively. Notice that in the preceding two equations, the gradients remain practically constant.

Thus, we see that, in spite of using the constitutive equations for the involved physical fluxes, it does not lead us to establish the expressions of $\beta_{q}$, $\beta_{\Pi }$ and $\beta_{J_{k}}$’s in terms of physical quantities.

#### 3.2.4. Rate of Change of Entropy in the Slow Domain of Time Using Constitutive Equations for the Physical Fluxes

In this domain of time, the evolution of the system is relatively slow. Therefore, the thermalization process remains comparable or faster than the processes under consideration. Moreover, the traditional discussion of the Boltzmann integro-differential equation, say, in terms of the Boltzmann H-function, applies in this domain of time (see for example [55,57,81]). The companion expression of entropy source strength to the Gibbs relation of Equation (Equation 25) (which indeed describes the evolution within the *slow domain of time*) on ignoring conservative body forces, reads as,
(95)$$\begin{array}{c} \begin{array}{c} \sigma_{s}=q\cdot \nabla \left(\frac{1}{T}\right)+\frac{1}{T}\Pi :\nabla u-\sum_{k}J_{k}\cdot \nabla \left(\frac{{\tilde{\mu }}_{k}}{T}\right)+\sum_{\alpha }\frac{{\tilde{A}}_{\alpha }}{T}\frac{d\xi_{\alpha }}{dt}>0 \end{array} \end{array}$$
which is the same expression that we have in Equation (Equation 26) when the body forces are ignored. The preceding expression also is expressed as,
(96)$$\begin{array}{c} \begin{array}{c} \sigma_{s}=-\frac{1}{T^{2}}\left(q-\sum_{k}{\tilde{\mu }}_{k}^{⦵}J_{k}\right)\cdot \nabla T+\frac{1}{T}\Pi :\nabla u-R\sum_{k}\frac{1}{c_{k}}J_{k}\cdot \nabla c_{k}+\sum_{\alpha }\frac{{\tilde{A}}_{\alpha }}{T}\frac{d\xi_{\alpha }}{dt}>0 \end{array} \end{array}$$
wherein we used the following transformations: (97)$$\begin{array}{c} \begin{array}{c} -\sum_{k}J_{k}\cdot \nabla \left(\frac{{\tilde{\mu }}_{k}}{T}\right)=\frac{1}{T^{2}}\sum_{k}{\tilde{\mu }}_{k}^{⦵}J_{k}\cdot \nabla T-R\sum_{k}\frac{1}{c_{k}}J_{k}\cdot \nabla c_{k} \end{array} \end{array}$$
and using the following expression of chemical potential, ${\tilde{\mu }}_{k}$ (based on the lines that we use in chemical thermodynamics [59,63,64,82]),
(98)$$\begin{array}{c} \begin{array}{c} {\tilde{\mu }}_{k}={\tilde{\mu }}_{k}^{⦵}+RT\ln c_{k} \end{array} \end{array}$$
where ${\tilde{\mu }}_{k}^{⦵}$ is the chemical potential in the standard state of unit concentration of the component *k* at the given magnitudes of physical fluxes to which ${\tilde{\mu }}_{k}$ belongs and *R* is the universal gas constant.

We now use the appropriate forms of constitutive equations of Section 3.2.1 in the *slow domain of time*.

(I).Since Equation (Equation 96) is the description within the segment belonging to the *slow domain of time* of the evolution, in general, we do have $\frac{dq}{dt}⟶0$, $\frac{d\Pi }{dt}⟶0$ and all $\frac{dJ_{k}}{dt}⟶0$. This also implies $\frac{d\nabla T}{dt}⟶0$, $\frac{d\nabla u}{dt}⟶0$ and all $\frac{d\nabla c_{k}}{dt}⟶0$.Therefore, the corresponding constitutive equations of Section 3.2.1 are transformed to
(99)$$\begin{array}{c} \begin{array}{c} q=-\lambda \nabla T,\;\Pi =2\eta \nabla u,\;J_{k}=-\rho D_{k}\nabla c_{k} \end{array} \end{array}$$
for Equations (Equation 67), (Equation 69), (Equation 70), (Equation 73), (Equation 74) and (Equation 75), that is, in the case of the *Maxwell–Cattaneo-type* and *Jeffreys-type* constitutive equations.(II).In the case of Guyer–Krumhansl’s relation, Equation (Equation 68), it reads in the slow domain of time as
(100)$$\begin{array}{c} \begin{array}{c} q=-\lambda \nabla T+l^{2}\nabla^{2}q+2l^{2}\nabla \left(\nabla \cdot q\right) \end{array} \end{array}$$(III).In the case of Giesekus upper convected and Leonov models for visco-elastic fluids, that is, Equations (Equation 71) and (Equation 72), we have the following operative expressions, respectively:
(101)$$\begin{array}{c} \begin{array}{c} \Pi =2\eta \nabla u-\frac{\alpha }{G}\Pi \cdot \Pi \end{array} \end{array}$$
and
(102)$$\begin{array}{c} \begin{array}{c} \Pi =2\eta \nabla u-\frac{1}{2G}\Pi \cdot \Pi . \end{array} \end{array}$$

Therefore, corresponding to the expressions of constitutive equations listed in (I), (II) and (III) above, we list below the expressions of entropy source strength, respectively:(i).Thus on using the expressions contained in Equation (Equation 99), the entropy source strength given in Equation (Equation 96) is transformed to
(103)$$\begin{array}{c} \begin{array}{c} \sigma_{s}=\frac{1}{\lambda T^{2}}\left(q-\sum_{k}{\tilde{\mu }}_{k}^{⦵}J_{k}\right)\cdot q+\frac{1}{2\eta T}\Pi^{2}+\frac{R}{\rho }\sum_{k}\frac{1}{D_{k}c_{k}}J_{k}^{2}+\sum_{\alpha }\frac{{\tilde{A}}_{\alpha }}{T}\frac{d\xi_{\alpha }}{dt}>0 \end{array} \end{array}$$(ii).In the case of rigid body heat conduction on using the expression of Equation (Equation 100), the entropy source strength reads as,
(104)$$\begin{array}{c} \begin{array}{c} \sigma_{s}=\frac{1}{\lambda T^{2}}q^{2}-\frac{l^{2}}{\lambda T^{2}}q\cdot \nabla^{2}q-\frac{2l^{2}}{\lambda T^{2}}q\cdot \nabla \left(\nabla \cdot q\right)>0 \end{array} \end{array}$$(iii).In the case of *visco-elastic fluids* on using the expressions of Equations (Equation 101) and (Equation 102), the respective expressions of entropy source strength that are obtained read as
(105)$$\begin{array}{c} \begin{array}{c} \sigma_{s}=\frac{1}{2\eta T}\Pi^{2}+\frac{\alpha }{2\eta TG}\Pi :\left(\Pi \cdot \Pi \right)>0 \end{array} \end{array}$$
and
(106)$$\begin{array}{c} \begin{array}{c} \sigma_{s}=\frac{1}{2\eta T}\Pi^{2}+\frac{1}{4\eta TG}\Pi :\left(\Pi \cdot \Pi \right)>0 \end{array} \end{array}$$

Notice that all expressions of entropy source strength in the *slow domain of time* consist of physical quantities amenable to experimental determination. Notice that there, we used the non-linear flux–force relationships and still the results are well within the LTE domain.

#### 3.2.5. The Rate of Change of Entropy Using Physical Fluxes as Additional Variables: Fast and Slow Domains of Time Taken Together

In the present Section 3.2.5 for the sake of better comprehending the discussion of Section 3.2.3 and Section 3.2.4, let us use the Gibbs relation of Equation (Equation 64) as it is, except we have dropped the rotation–vibration term for the sake of simplicity of discussion, which would still remain unaffected if we include this source of irreversibility.

Thus, we now substitute each one of the constitutive equations listed in Section 3.2.1 in the Gibbs relation of Equation (Equation 64) and the results are as listed below:On substituting *Maxwell–Cattaneo-type* constitutive equations, Equations (Equation 67), (Equation 70) and (Equation 74), the results are,
(107)$$\begin{array}{c} \begin{array}{c} \begin{array}{cc} \sigma_{s}= & \frac{\rho \beta_{q}}{\tau_{q}T}q^{2}+\frac{\rho \beta_{\Pi }}{\tau_{\Pi }T}\Pi^{2}+\frac{\rho }{T}\sum_{k}\frac{\beta_{J_{k}}}{\tau_{J_{k}}}J_{k}^{2}+\frac{1}{T^{2}}\sum_{k}{\tilde{\mu }}_{k}^{⦵}J_{k}\cdot \nabla T\\ \; & +\frac{1}{T}\left(\frac{\rho \lambda \beta_{q}}{\tau_{q}}-\frac{1}{T}\right)q\cdot \nabla T+\frac{1}{T}\left(1-\frac{2\eta \rho \beta_{\Pi }}{\tau_{\Pi }}\right)\Pi :\nabla u\\ \; & +\sum_{k}\left(\frac{\rho^{2}D_{k}\beta_{J_{k}}}{\tau_{J_{k}}T}-\frac{R}{c_{k}}\right)J_{k}\cdot \nabla c_{k}+\sum_{\alpha }\frac{{\tilde{A}}_{\alpha }}{T}\frac{d\xi_{\alpha }}{dt}>0 \end{array} \end{array} \end{array}$$
and the expression of the entropy flux density reads as,
(108)$$\begin{array}{c} \begin{array}{c} J_{s}=\frac{q-\sum_{k}J_{k}{\tilde{\mu }}_{k}}{T} \end{array} \end{array}$$We use Jeffreys’ type constitutive equations of Equations (Equation 69), (Equation 73) and (Equation 75), and assume the absence of body forces. Since no interaction with radiations is assumed, this exercise generates the following expression of entropy source strength:
(109)$$\begin{array}{c} \begin{array}{c} \begin{array}{cc} \sigma_{s}= & \frac{\rho \beta_{q}}{\tau_{q}T}q^{2}+\frac{\rho \beta_{\Pi }}{\tau_{\Pi }T}\Pi^{2}+\frac{\rho }{T}\sum_{k}\frac{\beta_{J_{k}}}{\tau_{J_{k}}}J_{k}^{2}+\frac{1}{T^{2}}\sum_{k}{\tilde{\mu }}_{k}^{⦵}J_{k}\cdot \nabla T\\ \; & +\frac{1}{T}\left(\frac{\rho \lambda \beta_{q}}{\tau_{q}}-\frac{1}{T}\right)q\cdot \nabla T+\frac{1}{T}\left(1-\frac{2\eta \rho \beta_{\Pi }}{\tau_{\Pi }}\right)\Pi :\nabla u\\ \; & +\sum_{k}\left(\frac{\rho^{2}D_{k}\beta_{J_{k}}}{\tau_{J_{k}}T}-\frac{R}{c_{k}}\right)J_{k}\cdot \nabla c_{k}+\sum_{\alpha }\frac{{\tilde{A}}_{\alpha }}{T}\frac{d\xi_{\alpha }}{dt}\\ \; & +\frac{\rho k\beta_{q}}{T}q\cdot \frac{d\nabla T}{dt}-\frac{2\rho \eta \beta_{\Pi }\tau_{\Pi }^{\prime }}{\tau_{\Pi }T}\Pi :\frac{d\nabla u}{dt}\\ \; & +\frac{\rho^{2}}{T}\sum_{k}\frac{\beta_{J_{k}}D_{k}\tau_{J_{k}}^{\prime }}{\tau_{J_{k}}}J_{k}\cdot \frac{d\nabla c_{k}}{dt}>0 \end{array} \end{array} \end{array}$$
and $J_{s}$ is still given by Equation (Equation 108). Notice that Equation (Equation 109) has three additional terms (the last three terms on its r.h.s.) compared to that of Equation (Equation 107).In the case of *rigid body heat conduction*, when we use the *Guyer–Krumhansl’s relation* of Equation (Equation 68) in the suitable version of the Gibbs relation Equation (Equation 64), the entropy source strength reads as,
(110)$$\begin{array}{c} \begin{array}{c} \begin{array}{c} \sigma_{s}=\frac{\rho \beta_{q}}{\tau_{q}T}q^{2}-\frac{\rho l^{2}\beta_{q}}{\tau_{q}T}q\cdot \nabla^{2}q-\frac{2\rho l^{2}\beta_{q}}{\tau_{q}T}q\cdot \nabla \left(\nabla \cdot q\right)\\ +\frac{1}{T^{2}}\left(\frac{\rho \lambda \beta_{q}T}{\tau_{q}}-1\right)q\cdot \nabla T>0 \end{array} \end{array} \end{array}$$
and the entropy flux density reads as,
(111)$$\begin{array}{c} \begin{array}{c} J_{s}=\frac{q}{T} \end{array} \end{array}$$In the case of *visco-elastic fluids*, when we use Equations (Equation 71) and (Equation 72) with the suitable version of the Gibbs relation of Equation (Equation 64), the respective outcome for the entropy source strength is,
(112)$$\begin{array}{c} \begin{array}{c} \sigma_{s}=\frac{1}{T}\left(1-\frac{2\rho \eta \beta_{\Pi }}{T\tau_{\Pi }}\right)\Pi :\nabla u+\frac{\rho \beta_{\Pi }}{T\tau_{\Pi }}\Pi^{2}+\frac{\rho \alpha \beta_{\Pi }}{GT\tau_{\Pi }}\Pi :\left(\Pi \cdot \Pi \right)>0 \end{array} \end{array}$$
and
(113)$$\begin{array}{c} \begin{array}{c} \sigma_{s}=\frac{1}{T}\left(1-\frac{2\rho \eta \beta_{\Pi }}{T\tau_{\Pi }}\right)\Pi :\nabla u+\frac{\rho \beta_{\Pi }}{T\tau_{\Pi }}\Pi^{2}+\frac{\rho \beta_{\Pi }}{2GT\tau_{\Pi }}\Pi :\left(\Pi \cdot \Pi \right)>0 \end{array} \end{array}$$

#### 3.2.6. Fast Versus Slow Domains of Time with Physical Fluxes as Additional Variables

Notice that in the description of Section 3.2.5, there is a possibility to arrive at the physical expressions of the parameters $\beta_{q}$, $\beta_{\Pi }$ and $\beta_{J_{k}}$’s. For example, consider Equation (Equation 107). There, we find three terms (i) the two in the second row and (ii) the first term in the third row of it. If we equate their coefficients to zero, the following expressions result:(114)$$\begin{array}{c} \begin{array}{c} \beta_{q}=\frac{\tau_{q}}{\rho \lambda T},\;\beta_{\Pi }=\frac{\tau_{\Pi }}{2\rho \eta },\;\beta_{J_{k}}=\frac{RT\tau_{J_{k}}}{\rho^{2}D_{k}c_{k}} \end{array} \end{array}$$

On using the expressions in Equation (Equation 114) in Equation (Equation 107), the result is Equation (Equation 103), which is a description in the *slow domain of time*. Similarly, (i) Equation (Equation 109) too gets reduced to Equation (Equation 103), (ii) Equation (Equation 110) to Equation (Equation 104), (iii) Equation (Equation 112) to Equation (Equation 105) and (iv) Equation (Equation 113) to Equation (Equation 106). Thus, the above transformations lead us directly to a description in *slow domain of time*. Therefore, the values of Equation (Equation 114) belong only to the *slow domain of time*. This outcome strongly substantiates two things. One is that our classification of thermodynamic variables into slow and fast types is a reality. Secondly the expressions of Equation (Equation 114) are yet another description that relegates the system to the slow domain of time.

It then implies that one cannot substitute the expressions of Equation (Equation 114) in any of the expressions of Section 3.2.3, which is a description of the *fast domain of time*.

At this juncture of our discussion, we only can say that the thermodynamics described in Section 3.2.3 needs further investigations to arrive at the physical expressions of the parameters $\beta_{q}$, $\beta_{\Pi }$ and $\beta_{J_{k}}$ applicable in the *fast domain of time*.

As far as the nonequilibrium steady states are concerned, it is clear from the above description of GPITT that there is no possibility to achieve it in the *fast domain of time*. Thus, all studies of nonequilibrium steady states belong to the *slow domain of time*. The operation of Onsager relations obviously belongs to the *slow domain of time*. Details of it in the GPITT framework will be discussed separately.

## 4. The Nature of Temperature in GPITT

In the entire development of GPITT, we used the same symbols for intensities *T* and *p*, whereas the chemical potentials, chemical affinities, affinity of collisional equilibration of the population of quantum states, other affinities of Section 3.1 and the quantum level composition variables are denoted as nonequilibrium quantities and represented by placing widetilde $\;\tilde{}\;$ over the symbols being used. To further illustrate why *T* and *p* do not need to be identified as nonequilibrium intensities, we discuss the case of temperature.

In equilibrium thermodynamics, temperature is legitimized using the zeroth and the second laws of thermodynamics, and it is coincided with the experimentally measured temperature on the Kelvin scale. Recall that, thanks to the advancement of techniques of ultrafast measurements, nowadays, we are able to register temperature from nanoseconds to even within a couple of picoseconds [83,84,85,86,87]. In spite of this, in such a short time scale, no significant amount of heat would flow from hot to cold parts of a system. In other words, the following standard thermodynamic inequality that determines the net heat gained or given out by the system reads as
(115)$$\begin{array}{c} \begin{array}{c} dQ_{1}\left(\frac{1}{T_{1}}-\frac{1}{T_{2}}\right)>0 \end{array} \end{array}$$

The minimum time of observations that would meet the requirement described in Equation (Equation 115) is generally asserted as the time duration in which each molecule of the local pocket undergoes at least one collision. This time duration would be more than a couple of nanoseconds, depending on the physical state of the system. In other words, it is proportional to the average kinetic energy of all the molecules of the tiny pocket interacting with the measuring gadget (see for example [88]). This does imply that by default, the said tiny mass element of the system is treated as a spatially uniform entity. This last mentioned fact is inherent in the prescription of LTE.

Recall the statement in Section 3.1 that very fast chemical reactions are first completed at the initial temperature. That is, the chemical conversion takes place at a constant temperature. The same analogy holds well for the temperature appearing in the Gibbs relation valid for the *fast domain of time*, Equations (Equation 76) and (Equation 85) and other thermodynamic equations those follow from these Gibbs relations, that is, *T* appearing in them corresponds to the initial state because the thermalization process is much slower than the relaxation time of the heat flux density, $\tau_{q}\approx 10^{-13}s$. However, presently, we have no clue to arrive at the expressions of the coefficients $\beta_{q}$, $\beta_{\Pi }$ and $\beta_{J_{k}}$ valid for the *fast domain of time*.

Further, independently establishing the legitimacy of *T* function led us to formulate a generalized zeroth law of thermodynamics [89] that reads as

“*Three tiny volume elements are instantly and simultaneously isolated from their respective nonequilibrium systems and in the same instant they are brought into diathermal contacts as closed rigid systems, 1 with 2 and 2 with 3. If within the short time interval of sensing of the thermal interactions it is found that 1 is in momentary thermal equilibrium with 2 and 2 is with 3 then 3 is also in momentary thermal equilibrium with 1. The momentary thermal equilibrium means that if the volume elements possessed the heat fluxes then they remain unaffected during the minimum short period of thermal interactions and if no heat flux existed, both such nonequilibrium and the equilibrium states included, no heat flux gets generated during the said diathermal contact. The making of a diathermal contact between the tiny volume elements one of which having a heat flux and the other one without it is not forbidden*”.

The temperature function provided by this version of the zeroth law of thermodynamics we opt to coincide it with the experimentally measured temperature on the standard Kelvin scale, which is often referred to as the local equilibrium temperature. Additionally, we can interpret the temperature function given by the generalized zeroth law of thermodynamics (in a sense) as the contact temperature conceptualized earlier [90,91,92]. The temperature function so arrived at corresponds to the *slow domain of time*. This is so because it meets the local level version of the condition of Equation (Equation 115). Obviously, for the *fast domain of time*, temperature may also remain virtually constant as described in Section 3.1.

## 5. Concluding Remarks

The main aim of this paper was to check whether the concept of LTE that has been used to develop CIT is as rigid as it is impressed upon by CIT. Therefore, only relevant aspects were discussed herein. The final conclusion is that the CIT missed recognizing the existence of certain internal irreversible processes, say, for example, that associated with the existence of physical fluxes. Actually, the existence of physical fluxes originates in the nonequilibrium population of the molecular translational quantum states. Hence, there does exist corresponding internal irrversible scalar processes determined by molecular collisions. This was accounted for by replacing the conventional composition variables $\left\{x_{k}\right\}$ by the quantum-level composition variables $\left\{{\tilde{x}}_{k,\;j}\right\}$. Since we replaced a set of composition variables by another set of composition variables, there is no reason to believe that by this act, LTE breaks down. This is because when we use the composition variables $\left\{x_{k}\right\}$ for a tiny mass element, the accompanying assumption is the spatial uniformity of it. This requirement by any stretch of imagination would not get violated just by replacing $\left\{x_{k}\right\}$ by $\left\{{\tilde{x}}_{k,\;j}\right\}$, as both are the composition variables, and the macroscopic-level composition variable $\left\{x_{k}\right\}$ and the quantum-level composition variables $x_{k,\;j}$ both do exist when the system is spatially uniform. When the system is spatially non-uniform, the existing composition variables are $\left\{x_{k}\right\}$ and $\left\{{\tilde{x}}_{k,\;j}\right\}$. With this understanding, the thermodynamic framework that is developed is known as GPITT. Further, it is interesting to see that the GPITT is easily transformed to a framework similar to TIV, which in turn also gets transformed to an EIT-type framework. That is, the incorporation of additional variables over and above the traditional ones does not imply the breakdown of LTE. Therefore, one needs to freshly investigate the question of what is meant by the breakdown of LTE by reexamining its present understanding. Perhaps the very recent definition of nonequilibrium temperature proposed in [51] may also help us to understand what is meant by the domain beyond LTE.

Thus, it gets demonstrated that the concept of LTE is more flexible than what the conventional CIT impresses upon. In other words, we have demonstrated in this presentation that the GPITT and its all variants for example, as EIT and TIV types may also be termed as “Local Thermodynamic Equilibrium with internal degrees of freedom”.

In Section 3.1 and Section 3.2, our discussion is based on the identification of fast and slow variables; hence, there exist two corresponding domains of time. The conventional variables ($u,\;v,\;\{x_{k}\}$) belong to the *slow domain of time*, whereas in the *fast domain of time*, it is an evolution under the condition of virtual isolation, and the operative variables are the additional thermodynamic variables. In the *fast domain of time*, though we do not have a direct equation of state for temperature (this is because there is no adequate time for the conduction mechanism of heat transfer to take place) as well as that for pressure, the indirect ones do exist (c.f. Equations (Equation 45) and (Equation 78)). It does not mean that the conventional concept of temperature is not applicable.

At this juncture of our discussion, let us recall the assertion of Bridgman [93,94],
“*The universe of the operation of thermodynamics is determined by the instrumental operations of the laboratory. Thus all the measurements that we are now capable of performing the thermodynamic measurements form a sub-group.*".
and
“*…in general the analysis of such systems will be furthered by the recognition of a new type of large scale thermodynamic parameters of state, namely the parameters of the state which can be measured but not controlled. Examples are the order-disorder rearrangements in mixed crystals, measurable by X-rays, and dislocations in a solid, measurable by the attenuation of supersonic vibrations. These parameters are measurable, but they are not controllable, which means that they are coupled to no external force variable which might provide the means of control. And not being coupled to a force variable, they cannot take part in mechanical work. Such a parameter of state, which enters into no term in the mechanical work, can be shown by simple analysis to be one which can take part only in irreversible changes*."

The first statement of Bridgman seemingly appears to restrict the gamut of thermodynamics. For example, it seemingly impresses that the very fast processes may not be covered by thermodynamics. However, his second assertion is relatively flexible, and we see that this conjecture is realized in GPITT as well as in the conventional EIT [24,25,26,27,28,29], TIV [30,31,32,33,34,35,36,37,38,39], Keizer’s version of nonequilibrium thermodynamics [49], and for that matter in the recent approach by Lucia [51,95]. It seems that EIT, TIV, Keizer’s version [40,41,42,43,44,45,46,47,48,49] and that of Lucia [51,95], are guided by the above stated first assertion of Bridgman that gets reflected by the use of nonequilibrium temperature and nonequilibrium pressure as distinctly different physical entities than the corresponding local equilibrium ones. This obviously led to an assertion that the corresponding entropy and chemical potential functions also are nonequilibrium ones distinctly different than the local equilibrium ones from the point of view of their physical contents. GPITT and its versions discussed in this paper are the descriptions well within the LTE domain, hence therein legitimately appear the local equilibrium intensities, temperature and pressure. Thus, we see that the inability to write an equation of state for temperature in the fast domain of time in the GPITT versions of the type EIT and TIV is not a drawback because there appears the local equilibrium *T* indirectly in the operative equations of states of Equations (Equation 45) and (Equation 78). Compare it with the fact that in equilibrium thermodynamics, we have dU=TdS−pdV for a closed system carried reversibly. Hence, there we have $T={\left(\partial U/\partial S\right)}_{V}$. Whereas under the reversible adiabatic condition, we have ${\left(dU\right)}_{adiabatic}=-pdV$, but then there is no way from it to write down the equation of state for *T*. However, under the reversible adiabatic condition, we never question the legitimacy of the intensity *T*. This is illustrated by the fact that there, we legitimately use adiabatic gas equations $pV^{{}^{\gamma }}=\mathrm{const}.,\;TV^{{}^{\gamma -1}}=\mathrm{const}.\;\mathrm{and}\;T^{{}^{\gamma }}p^{{}^{1-\gamma }}=\mathrm{const}.$ (where the heat capacity ratio γ reads as $\gamma =C_{p}/C_{V}$ with *C* as the respective heat capacities), that is, there is no doubt about the legitimate existence of *S* and *T* during reversible adiabatic condition. Therefore, it is needed to understand that in the LTE domain, by default, we have local equilibrium functions *s*, *u*, *T*, *p*, and so on, whether it is the case of the fast or slow domains of time.

Similarly, recall that the local level per unit mass expressions of G given in Equations (Equation 22), (Equation 24), (Equation 41) and (Equation 56) all provide us a way to quantify the irreversibility in chemical interactions. The first one quantifies it in terms of the nonequilibrium population of internal molecular quantum states. The other ones are equivalent expressions of the first one. Hence, though there also appear additional variables $\xi_{rv},\;q,\;\Pi ,\;\left\{J_{k}\right\}$, etc., they have their origin in the nonequilibrium population of molecular quantum levels. From this angle of visualization, the expressions of Equations (Equation 22), (Equation 24), (Equation 41) and (Equation 56) are the different ways to quantify the existing chemical interactions. Hence, we earlier termed it as the *additional facets of chemical interactions* [53]. That is why we once again assert that the irreversibility is all about the imbalances in chemical interactions. For comparison, but not exactly in the same way, there does exist an approach of Keizer wherein the existing imbalances in the elementary processes as the basic ingredient [49] are used. On the other hand, in the approach of Oláh et al. [50], the thermodynamic fluxes and forces are considered to be composed of the opposing contributions, and the nonequilibrium is due to the imbalances in them. Recall that $\left\{{\tilde{x}}_{k,\;j}\right\}$ are the composition variables whose variation $\left\{d_{i}{\tilde{x}}_{k,\;j}\right\}$ is composed of the well-known internal process chemical reactions and the other quantum-level irreversible processes described in the main text of this paper. In this sense, we stated that GPITT looks like an internal variable theory.

Now a word on the prevailing qualm in the thermodynamic literature about the thermodynamic description of irreversible processes that once again was spelled out in [96]. This state of affair originated because there we have various definitions of entropy function [97] right from the thermodynamic, Boltzmann, Gibbsian, information theoretic, Jayens and so on. Each definition has its own origin and range of applicability, and reconciliation among them is still a distant task, particularly with reference to dealing with irreversibility. Of course, this stand is sound-looking. In spite of the thermodynamics, researchers have developed a good number of irreversible thermodynamic frameworks, and each one of them was shown to have its own domain of applicability but still be lacking in unanimity among them. In this respect, we feel that the subject matter discussed in the present paper may turn out as a first step to lessen to some extent the gap among these thermodynamic frameworks of irreversible processes. Notice that in the present paper, we investigated a very crucial ingredient associated with LTE, that is, the spatial uniformity of the envisaged tiny mass elements. This leads us to bring within the LTE domain many thermodynamic frameworks which are being considered as descriptions beyond the LTE domain. Therefore, the results of this paper may be considered a first step taken that perhaps would prove in helping to resolve the above spelled-out ambiguities.

## Figures and Tables

**Figure 1 entropy-25-00145-f001:**
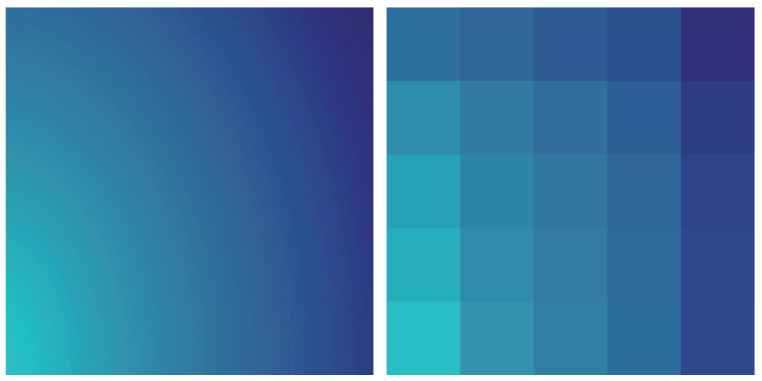
In the LTE (local thermodynamic equilibrium) approximation, a continuous gradient of, say, temperature in a system (left frame) is replaced by a number of small subsystems, each with its uniform temperature and pressure. Thus, the full system with its continuous irreversibility is represented by a collection of equilibrium systems with the irreversibility located at their boundaries, i.e., endoreversible systems (right frame) [8,9].

**Figure 4 entropy-25-00145-f004:**
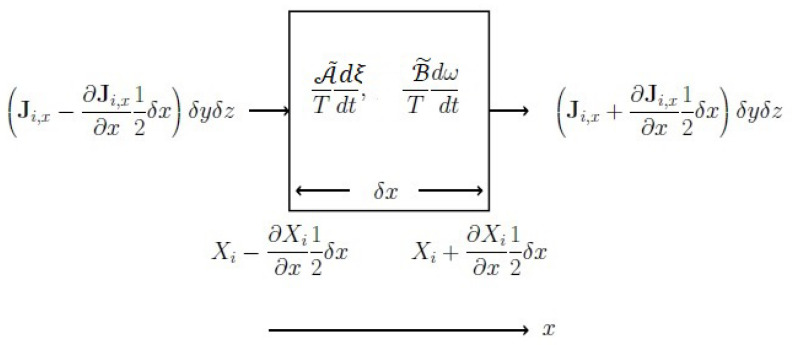
A schematic representation of the mathematical description in one dimension of the processes taking place within and across a tiny volume element of a non-uniform fluid and variation of intensive properties on account of existing gradients. $J_{i,x}$ is the flux density of the *i*-th process in the *x*-direction and $X_{i}$ is the *i*-th intensive property. The rest of the symbols are self explanatory; see Equation (Equation 17). For the sake of simplicity, we show only one flux density and only one intensive property, but by changing the subscript *i*, we include all relevant quantities. The mathematical expressions of the scalar sources of entropy production are described within the tiny volume element.

**Figure 5 entropy-25-00145-f005:**
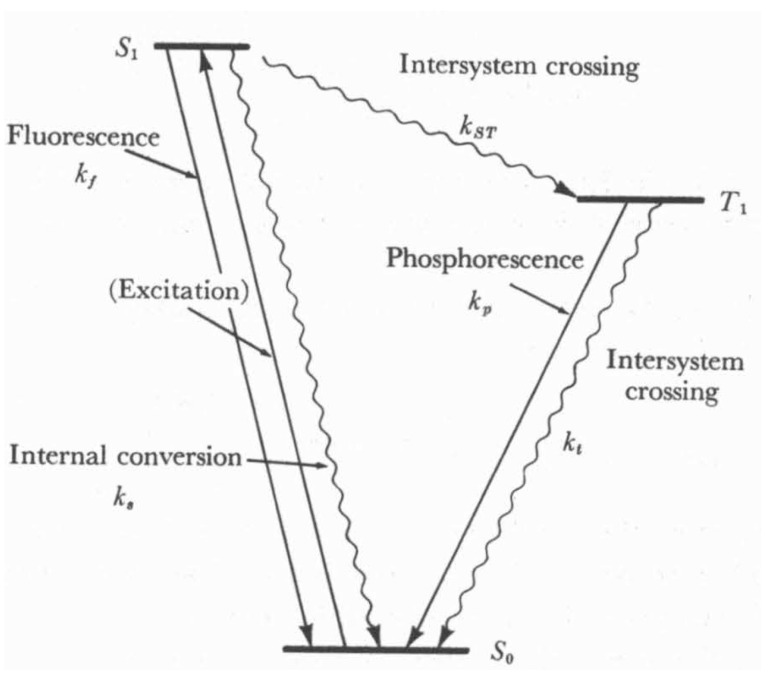
A schematic representation of the photo-physical processes, the Jablonski diagram for an organic molecule. It is a convention to represent the radiative transitions by straight arrows, ⟶, and the non-radiative ones by ⇝. $k_{ST}$ is the rate constant for the non-radiative intersystem crossing between the excited singlet ($S_{1}$) and the triplet state ($T_{1}$), $k_{f}$ is the rate constant of fluorescence emission from the singlet state ($S_{1}$), $k_{s}$ is the rate constant for internal conversion between singlet states (herein shown between $S_{1}$ and $S_{0}$) and $k_{t}$ is the rate constant for the non-radiative intersystem crossing from triplet ($T_{1}$) to ground singlet state $S_{0}$.

**Figure 6 entropy-25-00145-f006:**
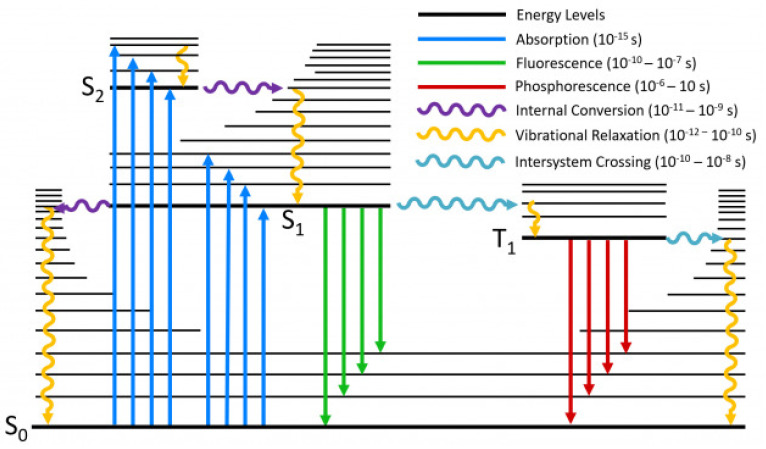
A schematic representation of the various electronic energy levels (singlet ground state $S_{0}$, first and second singlet excited states $S_{1}$ and $S_{2}$, and the first triplet excited state $T_{1}$) for an organic molecules and associated processes with corresponding time scales.

**Figure 7 entropy-25-00145-f007:**
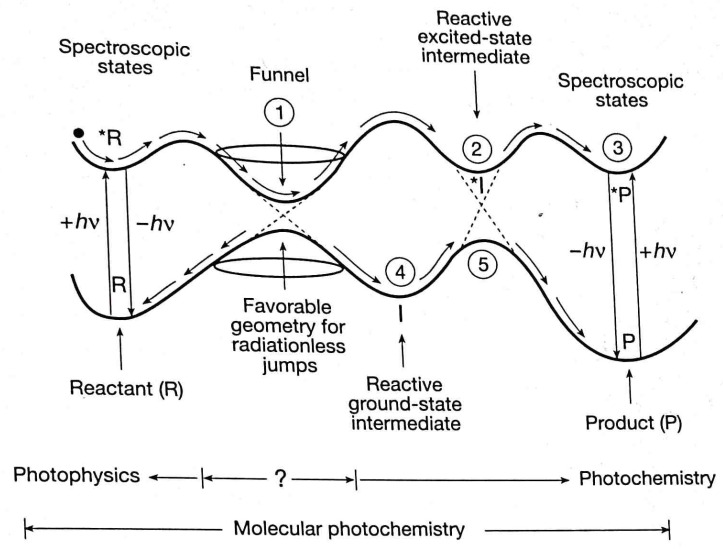
A representative potential energy surface for photochemical reaction R⟶P through the highly reactive intermediate I. The lower surface starts with the ground singlet state, $S_{0}$ and R⟶I⟶P is the pathway of thermal reaction. The upper surface starts with its excited singlet state, say, $S_{1}$ and describes the photochemical adiabatic pathway ${}^{*}R⟶{\;}^{*}I⟶{\;}^{*}P$ with excited state highly reactive intermediate ${}^{*}I$. For the sake of simplicity, the potential energy surfaces of R and ${}^{*}R$ as well as that of P and ${}^{*}P$ are assumed to be vertically similar. The "?" indicates a "twilight zone", where the distinction between photochemistry and photophysics is fuzzy. This region is termed as funnel, the region ➀ (this diagram was taken from [60]).

## Data Availability

Not Applicable.

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
