# Peer review of "How Flexible Is the Concept of Local Thermodynamic Equilibrium?"

_entropy, 2023, doi:10.3390/e25010145_

Round 1

Reviewer 1 Report

See attached text

Author Response

Added as an attachment.

Reviewer 2 Report

While I am technically unable to evaluate the scientific soundness of the paper, intuitively I think that this is an excellent paper with a most encouraging conclusion that thermodynamics of irreversible phenomena can be rigorously studied.

Qualm of that possibility has been raised and even the concept of irreversibility has been questioned. Jos Uffink in a 2001 article, "Bluff your way in the Second Law of Thermodynamics," has this to say:

This summary leads to the question whether it is fruitful to see irreversibility or time-asymmetry as the essence of the second law. Is it not more straightforward, in view of the unargued statements of Kelvin, the bold claims of Clausius and the strained attempts of Planck, to give up this idea? I believe that Ehrenfest Afanassjewa was right in her verdict that the discussion about the arrow of time as expressed in the second law of the thermodynamics is actually a red herring. The only way to evaluate such a proposal is by making up a balance-sheet. What would we lose and what would we gain? It is clear that in fact all concrete applications of the second law in classical thermodynamics, even in the work of the most outspoken proponents of the claim that this law implies universal irreversibility, are restricted to systems in equilibrium. This holds for Kelvin and Planck, but also more recent text books (e.g. (Becker 1967)). A general opinion among thermodynamicists is even that the theory is incapable of dealing with systems out of equilibrium; (see the quotation from Bridgman on page 3). Clearly, in terms of concrete applications, we would lose very little. What, then, do we gain with this proposal? The main advantage is, to my mind, that the second law would no longer represent an obstacle to the reconciliation of different theories of physics. More specifically, attempts to reduce thermodynamics to, or at least to harmonise it with, a mechanistic world picture would get a new lease of life.

I wonder whether the authors would care to comment on Uffink's assertion that "the theory is incapable of dealing with systems out of equilibrium."

Author Response

Added as an attachement.

Reviewer 3 Report

The paper develops an approach to the lcal equilibrium in non-equilibrium thermodynamics. This approach is justified by the relaxation time and the differential volume considered in deriving the equation.

I think that the approach suggested must be better explained in relation to the local fluctuations when the fluctuations are comparable with the measured quantities, which can appens in differential volume approach, as highlighted by Pauli in his textbooks. Morevoer, I suggest to consider the approach developed in relation to the non equilibrium one recently developed by Lucia et al. in relation to non equilibrium temperature, published in Materials. It could be interesting the comparison of the two approaches.

In line 481 th --> the

Author Response

Added as an attachment. 

Round 2
